



# Ground-based validation of the MetopA and B GOME-2 OClO measurements

Gaia Pinardi[1], Michel Van Roozendael[1], François Hendrick[1], Andreas Richter[2], Pieter Valks[3], Ramina Alwarda[4], Kristof Bognar[4*], Udo Frieß[5], José Granville[1], Myojeong Gu[6], Paul Johnston[7], Cristina Prados-Roman[8], Richard Querel[7], Kimberly Strong[4], Thomas Wagner[6], Folkard Wittrock[2], and Margarita Yela Gonzalez[8]

[1]Royal Belgian Institute for Space Aeronomy (BIRA-IASB), Av Circulaire 3, 1180 Uccle, Belgium
[2]Institute of Environmental Physics (IUPB), University of Bremen, Otto-Hahn-Allee 1, D-28359 Bremen, Germany
[3]Deutsches Zentrum für Luft-und Raumfahrt (DLR), Institut für Methodik der Fernerkundung (IMF), Münchener Str. 20, D-82234, Oberpfaffenhofen, Germany
[4]Department of Physics, University of Toronto, 60 St. George Street, Toronto, Ontario, M5S 1A7, Canada
[5]Institute of Environmental Physics (IUPH), University of Heidelberg, Im Neuenheimer Feld 229, Heidelberg, Germany
[6]Max-Planck-Institut für Chemie (MPIC), Hahn-Meitner-Weg 1, 55128 Mainz, Germany
[7]National Institute of Water and Atmospheric Research (NIWA), Private Bag 50061, Omakau, Central Otago, New Zealand
[8]Atmospheric Research and Instrumentation Branch, National Institute for Aerospace Technology (INTA), Madrid, 28850, Spain
[*]Now at: Institute of Space and Atmospheric Studies, University of Saskatchewan, Saskatoon, Saskatchewan, Canada

**Correspondence:** Gaia Pinardi (gaia.pinardi@aeronomie.be)

**Abstract.** This paper reports on ground-based validation of the atmospheric OClO data record produced in the framework of EUMETSAT's Satellite Application Facility on Atmospheric Chemistry Monitoring (AC SAF) using the GOME2-A and -B instruments over the 2007-2016 and 2013-2016 periods, respectively. OClO slant column densities are compared to correlative measurements collected from 9 NDACC Zenith-Scattered-Light DOAS (ZSL-DOAS) instruments distributed in both the

Arctic and Antarctic. Sensitivity tests are performed on the ground-based data to estimate the impact of the different OClO DOAS analysis settings. On this basis, we infer systematic uncertainties of about 25% between the different ground-based data analysis, reaching total uncertainties ranging from about 26% to 33% for the different stations.  Time-series at the different sites show good agreement between satellite and ground-based data, both for the inter-annual variability and the overall OClO seasonal behaviour. GOME-2A results are found to be nosier than those of GOME-2B, especially after 2011, probably due to

instrumental degradation effects. Daily linear regression analysis for OClO activated periods yield correlation coefficients of 0.8 for GOME-2A and 0.87 for GOME-2B, with slopes of 0.64 and 0.72, respectively. Biases are within 8 x$10^{13}$ molec/cm$^2$ with some differences between GOME-2A and GOME-2B, depending on the station. Overall, considering all the stations, a median bias of about -2.2 x$10^{13}$ molec/cm$^2$ is found for both GOME-2 instruments.



## 1 Introduction

The increase of the chlorine and bromine species in the stratosphere, due to the anthropogenic release of long-lived halogenated compounds, has led to dramatic ozone losses in the polar winter stratosphere starting in the eighties (e.g. Solomon et al., 1988, 1990; Solomon, 1999).

In polar regions, the chemical destruction of ozone is strongly influenced by the polar vortex, which results from the large-scale descent of cold air masses during winter. The polar vortex is also associated to strong Coriolis-related circumpolar

winds that prevent air mixing with lower latitudes. In the Northern Hemisphere, due to the inhomogeneous distribution of land masses, disturbances of the Arctic vortex by vertical propagation of planetary waves is frequent, while the Antarctic vortex usually remains stable and more or less symmetric until at least late spring (November).

During winter, temperatures inside the vortex can drop below the threshold for the formation of polar stratospheric clouds (PSCs), and heterogeneous reactions on PSC-particles convert ozone-inert chlorine reservoirs (mainly $ClONO_2$ and $HCl$) into

ozone destroying species (active chlorine, mainly $Cl$, $ClO$ and $ClOOCl$), see, e.g., Solomon (1999). This chlorine activation is the prerequisite for ozone destruction by catalytic cycles like the ClO–ClO and the ClO–BrO cycle (McElroy et al., 1986; Molina and Molina, 1987) after the return of sunlight in the polar spring. OClO is mostly created by the reaction between ClO and BrO (ClO + BrO –> OClO + Br) (Solomon et al., 1987; Toumi, 1994; Renard et al., 1997). OClO has a very short lifetime of a few seconds in the sunlit atmosphere due to its photolysis (OClO + h$\nu$ –> ClO + O), which prevents the build-up of

significant amounts until large solar zenith angles (SZAs) are reached. Nighttime and twilight OClO are thus a good indicator of chlorine activation (Sessler et al., 1995; Renard et al., 1997; Tørnkvist et al., 2002). Although OClO is only formed in sizeable quantities during the night, solar backscatter measurements of OClO columns can be performed from space near the terminator where the photolysis efficiency is reduced.

The emission of long-lived chlorine and bromine containing substances has been regulated since 1987 after the implementa-

tion of the Montreal Protocol and its amendments. As a result, atmospheric levels of the ozone-destroying precursor substances have decreased over the last decades. Monitoring of stratospheric chlorine and bromine contents remains important to assess the effectiveness of the regulatory measures taken, in particular in the context of climate change and its impact on ozone recovery.

Halogen oxides such as BrO and OClO can be measured using the Differential Optical Absorption Spectroscopy (DOAS)

method (Platt and Stutz, 2008) owing to their structured absorption cross-sections in the UV and visible parts of the spectrum. For OClO, the first detection from the ground was reported by Solomon et al. (1987) in Antarctica, and subsequently by many other measurements in both hemispheres (Solomon et al., 1988, 1990; Gil et al., 1996; Kreher et al., 1996; Otten et al., 1998; Richter et al., 1999; Tørnkvist et al., 2002; Vandaele et al., 2005; Frieß et al., 2005). Observations from aircraft (Schiller et al., 1990) and from balloons (Pommereau and Piquard, 1994; Renard et al., 1997) followed.

The first OClO retrievals from nadir satellite data were performed using the Global Ozone Monitoring Experiment (GOME) by Wagner et al. (2001, 2002); Burrows et al. (1999); Kühl et al. (2004) and Richter et al. (2005). This was followed by measurements from the Scanning Imaging Spectrometer for Atmospheric Chartography (SCIAMACHY, Kühl et al. (2006)), the





Ozone Monitoring Instrument (OMI, OMOCLOv3), GOME-2 (Richter et al., 2015; Valks et al., 2019a, b), and the TROPO-spheric Monitoring Instrument (TROPOMI, Meier et al. (2020); Puķīte et al. (2021a, b)).

Richter et al. (2015) illustrated the possibility to retrieve consistent dataset of OClO slant column densities (SCDs) from both GOME-2A and GOME-2B sensors. Settings proposed by Richter et al. (2015) study were implemented at DLR for the AC SAF data products (Hassinen et al., 2016) within the GOME Data processor (GDP) 4.8 (Valks et al., 2019a, b) for the period 2007 to 2016, and are under focus in this study.

These global long-term nadir satellite datasets offer interesting perspectives to study inter-hemispheric and inter-annual
differences in the activation of halogens, their dependence on meteorological parameters and their long-term trends. To allow for reliable exploitation of the long time-series (starting in 1995 with GOME), it is essential to validate the different data sets. At present, to our knowledge, only a small number of studies quantitatively intercompared OClO datasets, and mostly on a few seasons/episodes/years (Oetjen et al., 2011; Richter et al., 2015; Kühl et al., 2006; Puķīte et al., 2021a, b).

In this paper we present a validation approach focusing on polar regions, by addressing the quality of the GOME-2A and
GOME-2B OClO ACSAF data records over 8 stations, during the time-period from 2007 until 2016. The satellite slant columns are compared to correlative observations acquired by independent ground-based DOAS spectrometers in zenith-sky geometry and the results for both satellites are compared and discussed. The paper is organized as follows: Section 2 presents the OClO algorithm applied to GOME-2, while Sect. 3 presents the ground-based ZSL-DOAS datasets and the comparisons method. The validation results are discussed in Sect. 4 and conclusions are given in Sect. 5.

## 65   2   GOME-2 OClO data

The second Global Ozone Monitoring Instrument (GOME-2) is a nadir-looking UV-visible spectrometer measuring the solar radiation backscattered by the atmosphere and reflected by the Earth surface and clouds in the 240–790 nm wavelength interval at a spectral resolution of 0.2–0.5 nm full width at half maximum (FWHM) (Munro et al., 2016). There are three GOME-2 instruments flying on Sun-synchronous polar orbits on board the Meteorological Operational satellites (MetOp-A, MetOp-B
and MetOp-C, launched in October 2006, September 2012, and November 2018, respectively). They have an Equator crossing time of 09:00-09:30 local time in the descending node. The default swath width of the GOME-2 across-track scan is 1920 km, allowing global Earth coverage within 1.5–3 days at the Equator, with a nominal ground pixel size of 80×40km$^2$. Since 15 July 2013, GOME-2A is measuring on a reduced swath mode of 960km, with a ground pixel size of 40×40km$^2$.

Following the initial study of Richter et al. (2009), an improved OClO slant column retrieval algorithm was developed for
both GOME-2A and –B in the framework of an AC SAF Visiting Scientist project (Richter et al., 2015). This led to a clear improvement compared to earlier results. The settings, summarized in Table 1, were implemented by DLR in the AC SAF product portfolio as GDP 4.8 data records for GOME-2A (2007-2016) and GOME-2B (2012-2016). These data products can be found on the acsaf.eoc.dlr.de FTP server.





The GOME-2 GDP 4.8 OClO retrieval algorithm is fully described in the corresponding Algorithm Theoretical Basis Doc-
ument (Valks et al., 2019a) and detailed information about the development of the analysis can be found in Richter et al.
(2015).

The DOAS retrieval is performed in the UV wavelength range 345-389nm which was found to minimise both bias and
noise in retrieved OClO slant columns. The fit includes $NO_2$, $O_3$, $O_2$-$O_2$ and the Ring effect (see Table 1). The GOME-2
key data parameter Eta is included as another effective cross-section to correct for residual polarization errors in the level-1
product. This inclusion significantly improves the OClO fitting residuals. Two empirical correction functions (derived from
mean DOAS-fit residuals) are also included as additional (pseudo-) absorption cross-sections in the DOAS-fit: a mean residual
and a scan angle correction function. These two empirical functions correct for positive offsets and scan angle dependencies
in the OClO columns. Remaining biases in the OClO columns (e.g. non-zero OClO columns over areas without chlorine
activation), with temporal drifts observed mainly in the OClO data from GOME-2A (see Richter et al. (2015)), need to be
treated using an additional offset correction. A simple normalization is thus applied on an orbital basis. The mean OClO slant
column for the area between 50°N and 50°S (a latitude region without chlorine activation) is determined for each GOME-2
orbit,and subtracted from the retrieved OClO slant columns for the complete orbit, leading to normalized OClO slant columns
(SCD).

**Table 1.** DOAS settings used for the GOME-2 OClO retrieval in GDP 4.8.

| Variable | Detail |
|---|---|
| Fitting interval | 345-389 nm |
| Sun reference | Sun irradiance for GOME-2 L1 product |
| Wavelength calibration | Calibration of sun reference optimized by NLLS adjustment on convolved Chance and Spurr solar lines atlas |
| Polynomial | 4th order, 5 coefficients |
| Offset | linear |
| Absorption cross-sections: | |
| - OClO | Kromminga et al. (2003)(213K) |
| - $NO_2$ | Gür et al. (2005) (223K) |
| - $O_3$ | Gür et al. (2005) (223K and 243K) |
| - $O_2$-$O_2$ | Hermans et al. (1999) |
| - Ring effect | Vountas et al. (1998) |
| - Key data | Eta |
| - Empirical functions | mean residual and scan angle correction |

An illustration of OClO SCD maps for the Arctic in February 2011 and the Antarctic in August 2015 is given in Fig. 1.





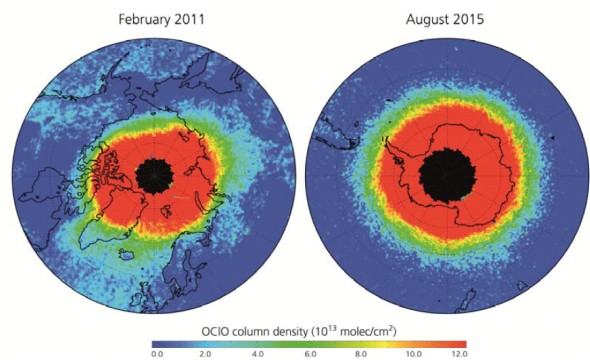

**Figure 1.** GOME-2 OClO maps for February 2011 and August 2015.

As OClO photolyses rapidly, it can only be observed at large solar zenith angle close to the terminator. Under these circumstances the calculation of an AMF and a vertical column is not trivial. It is complicated by rapid photolysis, the change in SZA along the line of sight, and also the uncertainty in the OClO vertical profile (Richter et al., 2005; Oetjen et al., 2011). Therefore, as done in previous studies, the GOME-2 GDP data product only contains (normalized) OClO slant columns densities (SCD).

A flag indicates when valid (enhanced) OClO column values can be expected from the GOME-2 data. The OClO flag is 100 set to 1 for daylight measurements with large solar zenith angle (85 º < SZA < 89º) and it is set to 2 for measurement during twilight (89 º < SZA < 92º), see (Valks et al., 2019b).

Figure 2 illustrates the GOME-2A and B datasets, by presenting the daily 90° SZA OClO SCD averages of both instruments, separated by hemisphere. As expected, OClO levels in the Southern hemisphere are usually larger than in the Northern hemisphere, and the year-to-year variability is larger in the latter. E.g., lower chlorine activation levels are found in 2009 and 2013 105 in the Northern hemisphere compared to other years. Outside the chlorine activation period, values should be very close to 0 in both hemispheres. This is usually the case in the first years of measurements of each instrument, although some negative or positive offsets (of up to 4 to 5 x$10^{13}$ molec/cm$^2$) appear for some of the years (e.g. 2010 in the Northern hemisphere or 2011, 2012 and 2013 for southern hemisphere for GOME-2A). These results suggest that there is still room for improvement in the current GOME-2 analysis.

**3  Comparison data and method**

**3.1  Ground-based NDACC ZSL-DOAS data**

As stated in the introduction, OClO columns have been retrieved from the ground since 1986 using the DOAS technique. For this study we selected 8 stations operating Zenith-Scattered sun Light (ZSL)-DOAS UV-Visible spectrometers from the Network for the Detection of Atmospheric Composition Change (NDACC, https://www.ndaccdemo.org/, last access on 28 115 June 2021), located above 60° latitude in both hemispheres and performing OClO SCD data retrievals. The geographical





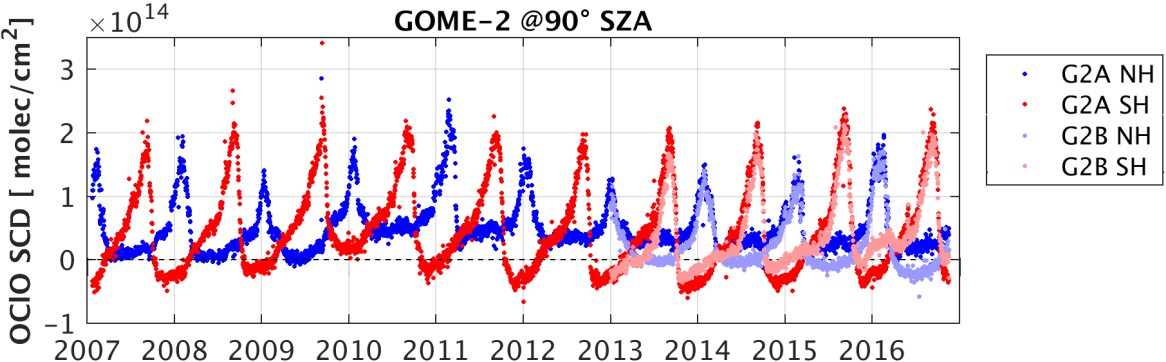

**Figure 2.** Daily GOME-2 OClO SCD time series for SZA = 90 $\pm 1°$.

distribution of these instruments is represented in Fig. 3 and a more extensive descriptions of the sites is given in Annex A1. This dataset provides a good temporal coverage, some of the stations reporting observations over the whole Metop-A operation period (2007-2016). A good coverage of the Arctic and Antarctic region is also achieved, with half of the stations in the Northern Hemisphere and half in the Southern Hemisphere. This ensemble of stations was also recently used for the validation

of TROPOMI OClO SCDs (Pinardi et al., 2020).

Specific details on the OClO SCD analysis are given in Table 2. As further described in Sect. 3.2, ground-based measurements are extracted at the solar zenith angle of the recorded GOME-2 pixels, for optimal photochemical coincidence with satellite observations. A fixed reference spectrum selected outside of the activated vortex period ensures that no OClO contribution comes from the reference, providing in this way absolute slant columns. For the UToronto instrument in Eureka, some

instrumental instabilities prevented the use of one yearly fixed spectrum for the analysis of some of the years, leading to a reduced temporal coverage of the comparisons (see Fig. 13 and 14).

From Table 2 it is clear that the ensemble of ground-based datasets is an aggregate of existing measurements and there is no harmonization in the retrieval choices of the different groups processing the OClO data. Different wavelength regions were used by each group for the OClO analysis, depending mainly on the spectral range covered by the respective instruments (see

Table A1 for the instrumental details). In most cases, retrievals were performed in the UV region between 345 and 392nm. One exception is NIWA who analysed its data in the visible spectral range (404-425nm, Kreher et al. (1996)). An illustration of the different OClO bands used in the different intervals is presented in Figure 4.

Another important difference is related to the OClO cross-section used, and its temperature. It can be seen that most of the groups use the Kromminga et al. (2003) cross-sections, while IUPB adopted the Kromminga et al. (1999) and UToronto the

Wahner et al. (1987) dataset at 204K. Moreover, within groups having adopted the Kromminga et al. (2003) data, most of them used the 213K dataset, while INTA and IUPH used the 233K dataset.





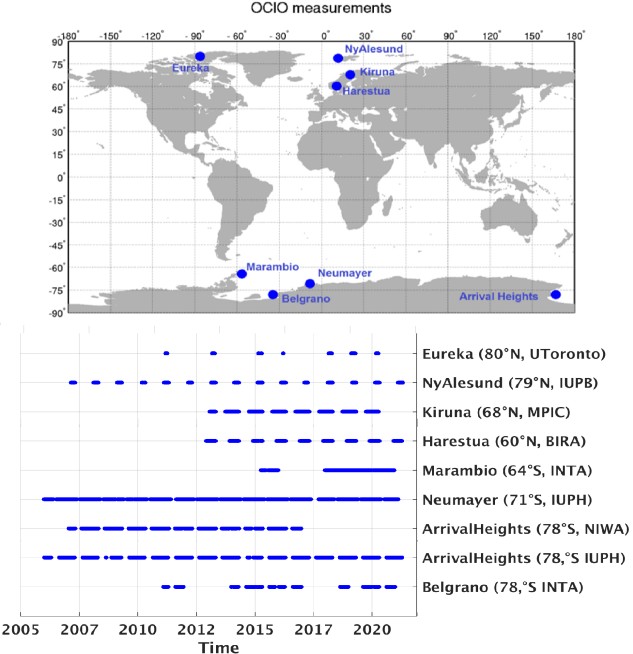

**Figure 3.** Geographical distribution and measurement time-periods of the UV-visible NDACC ZSL-DOAS instruments providing the correlative OClO measurements.

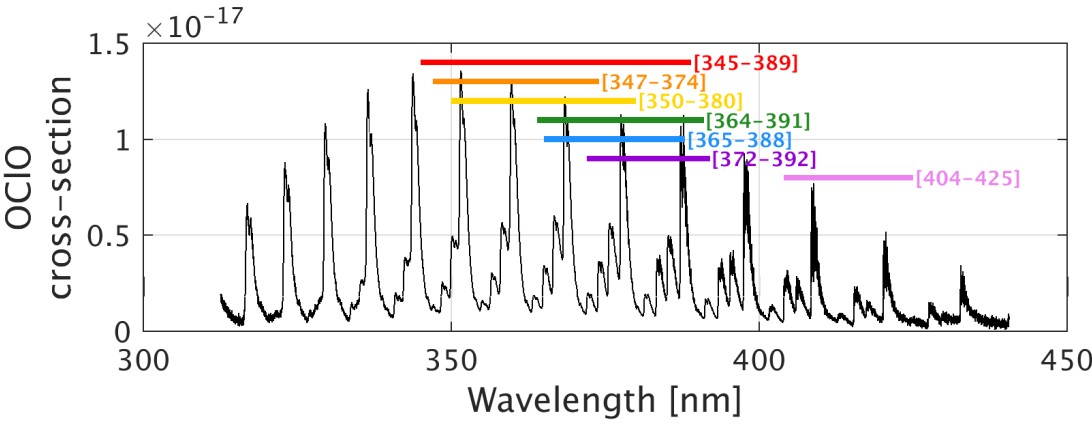

**Figure 4.** OClO absorption cross-section at 213K from Kromminga et al. (2003) and the different DOAS intervals used in this study.

Depending on the selected DOAS interval, the different groups include in their DOAS fit several other trace gas cross-sections ($NO_2$, $O_3$, $BrO$, $O_4$) in addition to OClO. Also they treat the Ring effect as a pseudo-absorber. Not all the absorbers





**Table 2.** Description of the different ground-based OClO datasets used in this study.

| Group | Station | Coordinates | wavelength range (nm) | Cross-sections | | | | | |
|---|---|---|---|---|---|---|---|---|---|
| | | | | OClO | NO$_2$ | O$_3$ | BrO | O$_4$ | Others |
| UToronto | Eureka | 80.05°N, 86.42°W | 350-380 | [a] **(204K)** | [d] (220K) | [e] (223K) | [h] (223K) | [j] (296K) | Ring[n] |
| IUPB | Ny-Ålesund | 78.9°N, 11.9°E | 365-388 | [b] (213K) | [d] (220K)* | - | - | [l] (298K) | Ring[o] |
| MPIC | Kiruna | 67.8°N, 20.4°E | 372-392 | [c] (213K), OClO×λ | [d] (220K) | [e] (223K) | - | [m] (273K) | Ring (213K, 263K) $Ring_1 \times \lambda^4$, $Ring_2 \times \lambda^4$ |
| BIRA | Harestua | 60.2°N, 10.7°E | 347-374 | [c] (213K) | [d] (220K) | [f] (223K, 243K)* | [h] (223K) | [m] (293K) | Ring[p] |
| INTA | Belgrano | 77.9°S, 34.6°W | 345-389 | [c] **(233K)** | [d] (220K)* | [f] (223K,243K)*+ | [h] (223K) | [m] (293K) | Ring[o] (250K) |
| | Marambio | 64.3°S, 56.7°W | | | | | | | |
| IUPH | Neumayer | 70.6°S, 8.3°W | 364-391 | [c] **(233K)** | [d] (220K, 298K) | [e] (223K, 293K) | [i] (228K) | [l] (298K) | Ring[n] |
| | Arrival Heights | 77.8°S, 166.6°W | | | | | | | $Ring \times \lambda^4$ |
| NIWA | Arrival Heights | 77.8°S, 166.6°W | 404-425 | [c] (213K) | [d] (220K) | [g] (218K) | - | - | Ring and H$_2$0 |

[a]:Wahner et al. (1987); [b]:Kromminga et al. (1999); [c]:Kromminga et al. (2003); [d]:Vandaele et al. (1998); [e]:Bogumil et al. (2003); [f]:Serdyuchenko et al. (2014); [g]:Brion et al. (1998); [h]:Fleischmann et al. (2004); [i]:Wilmouth et al. (1999); [j]:Greenblatt et al. (1990); [k]:Hermans et al. (2003); [l]:Hermans et al. (1999); [m]:Thalman and Volkamer (2013); [n]:Chance and Spurr (1997); [o]:QDOAS high resolution based on SAO: Chance and Kurucz (2010); [p]:SCIATRAN
*: I0 correction (Aliwell et al., 2002); +: with Puķīte et al. (2010) approach

are necessarily needed especially when a small wavelength interval is considered. E.g., the Ny-Ålesund IUPB analysis (365-388nm) does not include O$_3$ and BrO while the Kiruna MPIC analysis (372-392nm) does not include BrO. For the NIWA visible interval these 2 gases are also not necessary, while the water vapor cross-section should be considered.

In order to assess the uncertainties related to the use of different OClO DOAS fit settings by the different groups, we performed a series of sensitivity tests that are reported in the next subsection.

### 3.1.1 SCD error estimation

In this section we summarize the ground-based SCD error estimation. The random component of the uncertainty is evaluated using results from DOAS retrievals performed by each group, and, for the systematic uncertainty, we perform sensitivity tests to evaluate the impact of applying different retrieval settings, as presented in Table 2. The details of the different sensitivity tests are presented in Annex A2, and the results are summarized here and in the different tables.

*Random errors:*

Random errors on SCDs are estimated by each group as part of their DOAS analysis. As summarized in Table 3, median values for the different datasets range from 6 to 22% for SCD values of about $15\pm2$ x10$^{13}$ molec/cm$^2$ (representative of OClO measurements in activated conditions and median values of the SZA in between 86° to 90°, depending on the station). These values are globally consistent with past literature estimations (about 2 x10$^{13}$ molec/cm$^2$ for Neumayer and Arrival Heights



(Frieß et al., 2005), 4-10% at 90° SZA for the NIWA Arrival Heights (Kreher et al., 1996) and 20% for Ny-Ålesund data at 90° SZA (Oetjen et al., 2011)).

*Systematic errors:*

160 Systematic errors on OClO SCDs are estimated based on sensitivity tests performed using spectra recorded with the IUPB instrument in Ny-Ålesund during of a few days in February 2014. As presented in Annex A2, we investigated the impact of main differences that can be identified in Table 2, i.e, first, the choice of the OClO cross-sections source and its temperature, and secondly the different wavelength ranges.

The estimated systematic errors range between 2 and 15% for the uncertainty related to the OClO cross-section (see Fig. 165 A1) and a total uncertainty of about17% (Table A2). The values corresponding to each group's choice are indicated in the first column of the systematic uncertainty contributions in Table 3.

The errors due to the different group's retrieval choices are estimated through regression analysis of each setting with respect to the median OClO SCD values (see Fig. A2). The results present compact regression with RMS generally smaller than $2 \times 10^{13}$ molec/cm$^2$, except for IUPH and MPIC. As discussed in Annex A2, results for the latter two cases are likely 170 biased due to the limited wavelength range (up to about 390.4nm) of the Ny-Ålesund spectra. All intercepts except for IUPH are smaller than $1 \times 10^{13}$ molec/cm$^2$ (see Fig. A2), meaning that the observed bias is mostly multiplicative. The values corresponding to each group's choice are indicated in the second column of the systematic uncertainty contributions in Table 3. The largest impact on the slope is obtained for the MPIC and for UToronto cases, leading to a difference between all cases of about 18.5% (see Table A2). This value is considered as the maximum systematic uncertainty on the retrieval choice for 175 the systematic uncertainty contribution in Table A2, leading to a total maximum systematic uncertainty of about 25% when adding the contribution related to the OClO cross-section source.

*Expected systematic bias against GOME-2:*

The expected systematic bias due to differences between each group's analysis and the GOME-2 OClO retrieval settings is 180 investigated in a third test where the median OClO SCDs are replaced by the SCDs obtained using the GOME-2 settings in the scatter plots (see Fig. A3). For each group, the total expected systematic bias on OClO SCD consists of a first component due to the difference in the used OClO cross-section compared to Kromminga et al. (2003) (reported as the first number of the last column of Table 3) and a second component coming from the impact of other settings, as obtained in Fig. A3. The total expected systematic bias on OClO SCDs with respect to GOME-2 analysis ranges between 4% and 16% for the different 185 stations.

### 3.1.2 SCD offset correction

Although OClO SCD measurements used in this study are obtained using a fixed reference spectrum selected outside of the activated period to make sure that no residual OClO is contained in this reference, OClO SCD offsets are often observed in actual measurements due to instrumental effects leading to systematic spectral interferences with OClO absorption structures





**Table 3.** Error estimates for the different OClO analysis at each station, in percent. The random uncertainty is estimated from the DOAS fit uncertainty for an OClO SCD of $15\pm2$ x$10^{13}$ molec/cm$^2$. The systematic uncertainty is evaluated considering the impact of using different OClO cross-sections as well as different retrieval settings (see text and Fig. A1, A2, Table A2). The total uncertainty is calculated as the quadrature sum of random and systematic contributions. Estimation of the expected systematic bias with respect to the GOME-2 analysis setting is given in the last column (see text and Fig. A3).

| | Uncertainties [%] | | | Syst. Biases [%] |
|---|---|---|---|---|
| | Rand. | Syst. | Total | wrt GOME-2 |
| Stations | DOAS fit | | | (OClO xs; others choices: Tot) |
| Belgrano | 13 | 2; 0.2 | 13.1 | 2; 5: 5.4 |
| Arrival Heights (NIWA) | 22 | -; n.a. | n.a. | -; n.a.: n.a. |
| Arrival Heights | 15 | 2; 11 | 18.7 | 2; 3: 3.6 |
| Neumayer | 14 | 2; 11 | 17.9 | 2; 3: 3.6 |
| Marambio | 13 | 2; 0.2 | 13.1 | 2; 5: 5.4 |
| Harestua | 6.5 | -;4.5 | 7.9 | -;9: 9 |
| Kiruna | 22 | -;7.5 | 23.2 | -; 16: 16 |
| Ny-Ålesund | 10 | 2.5; 2.5 | 10.7 | 2.5;8: 8.4 |
| Eureka | 10 | 15; 4 | 18.5 | 15;1.3: 15.1 |

(e.g. thermal instabilities leading to changes in instrumental spectral response) or due to possible unknown atmospheric effects interfering with the OClO retrieval.

Such effects generally lead a systematic bias on the retrieved OClO SCDs that can vary in time, but usually with a time constant that exceeds the duration of a twilight period.

To further mitigate the impact of such biases, an empirical correction was designed and systematically applied to the ground-
based data sets.

The principle of this correction relies on the assumption that OClO bias sources are constant during a twilight period and therefore lead to an offset on the retrieved OClO SCDs. For each morning and evening twilight, we draw a Langley plot, i.e. a plot of the SCDs reported as a function of the OClO air mass factor (AMF). One example of such a plot is represented in Fig. 5, for the Harestua station on 13 January 2013. The AMF used for this purpose was empirically estimated from observed
OClO SCDs recorded during a series of chlorine activation events of various strengths (see Fig. 6). The AMF is here defined as the ratio of the measured slant column to the vertical column estimated at 70° of SZA, assuming that at this solar elevation a simple geometrical AMF can be used. The grey area in Fig. 6 indicates the range of the measured OClO AMFs, while the blue and green curves show AMFs calculated using the DISORT radiative transfer model coupled PSC-Box and initialized with SLIMCAT 3D-Chemical transport model simulations, as explained in Hendrick et al. (2007). The red line represents the
median value of the measured AMFs, which was used an input for the present analysis.





As can be seen in Fig. 5, a linear relationship is obtained between the empirical AMFs and the measured SCDs over a large range of SZA values. We also note that although the reference spectrum used to analyse these data was recorded well outside the activated period (in late April in this case) and therefore does not contain any sizeable OClO amount, the observed SCDs present an offset, i.e. the measured SCDs do not converge to zero for low AMF values. This offset is necessarily an artefact

and should be removed to restore physically consistent SCD values.

It must be noted here that this approach is only applicable for observations covering a sufficiently large range of SZAs. The limit on the minimum solar zenith angle has been empirically set to 86°. For high latitude observations during polar night conditions, when the SZA constantly exceeds 86°, an estimate of the offset was obtained by fitting a polynomial function to offsets derived during the illuminated periods.

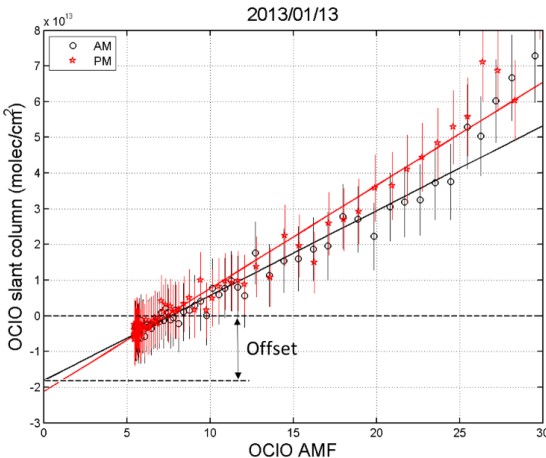

**Figure 5.** Illustration of the Langley plot method used to estimate offset artefacts on OClO SCD measurements. This case was obtained in Harestua on 13 January 2013.

Despite its empirical nature, this offset correction, which was derived independently for morning and evening data on each day, can be considered as objective as a) it is not linked to the satellite data and b) it is not based on subjective criteria such as the smoothness of the OClO timeseries.

This correction was applied to all ground-based datasets used in this study, except for NIWA measurements in Arrival Heights. At this site the method could not be used due to the unavailability of daily sequences of OClO measurements covering

a suitable range of SZAs.

Figure 7 presents an illustration of the impact of the correction for the Neumayer ground-based dataset time-series. The original data is displayed in light grey and the corrected one in black. The same data set is also represented as a function of the SZA in the lower panel. As can be seen, in this case, the main impact of the offset correction is to reduce the apparent noise on the low values of the OClO SCD. During periods of strong activations, changes are generally minor.





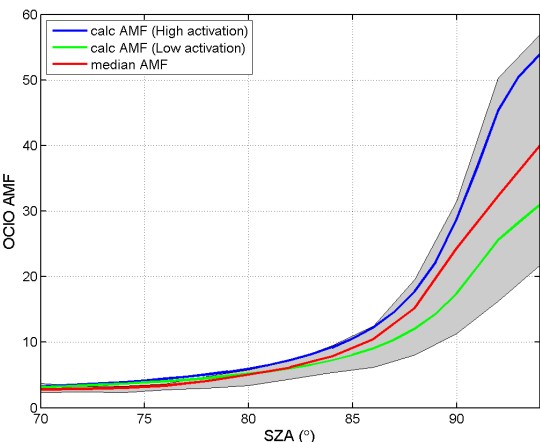

**Figure 6.** Illustration of the AMFs used for the Langley plots. The grey area indicates the range of the measured OClO AMFs, the red curve their median value, while the blue and green curves are AMFs calculated using the DISORT radiative transfer model coupled PSC-Box and initialized with SLIMCAT 3D-Chemical transport model simulations.

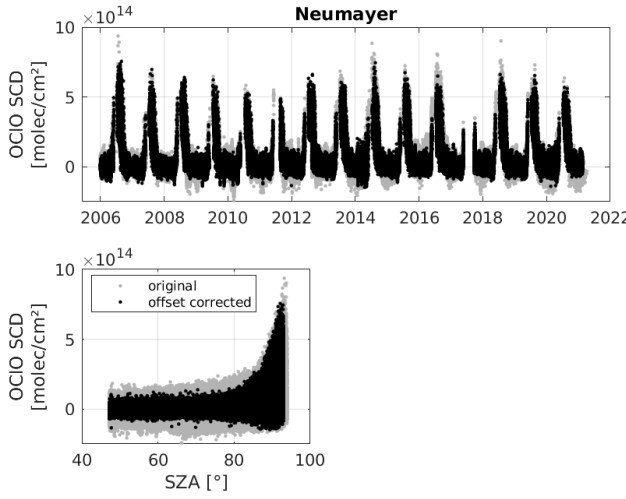

**Figure 7.** Illustration of the offsset correction impact on of Neumayer data (a) the time-series and (b) the SZA dependence.

## 3.2 Comparison method

For the comparison of GOME-2 and ZSL-DOAS data, a method similar to Richter et al. (2015) and Oetjen et al. (2011) was adopted. The GOME-2 GDP 4.8 OClO SCD data are extracted within 200 km of the different stations listed in Table 2. The mean value of the valid OClO SCD (oclo_flag value set to 1 or 2, i.e. between 85°and 92° SZA, Valks et al. (2019a, b)) is then





calculated for each day, in order to improve the signal to noise ratio. Coincidences are obtained by selecting ground-based data

that are within ±1° SZA of the mean daily satellite value. Error weighted averages are performed using provided ground-based and satellite errors.

    Comparisons of the daily coincidences are performed at each station for the whole available time-series. It should be noted that there is a non-constant number of points at SZA>85° throughout the year at some stations. This is even more the case after the reduced swath configuration was adopted for GOME-2A in July 2013. During several periods of the year (depending on

the location) no valid OClO SCD can be found and such periods tend to be longer after 2013.

    The approach of comparing slant columns (instead of vertical columns) relies on the assumption that satellite nadir and ground-based zenith sky light paths are comparable at large SZA (Oetjen et al., 2011). In other words, satellite AMFs (AMF-sat_nadir) and ground-based AMFs (AMFgb_zenith) are assumed to be similar. Oetjen et al. (2011) et al. calculated differences of up to 4% for the two observation geometries between 89° and 91° SZA and of 13% at 80° SZA in Ny-Ålesund.

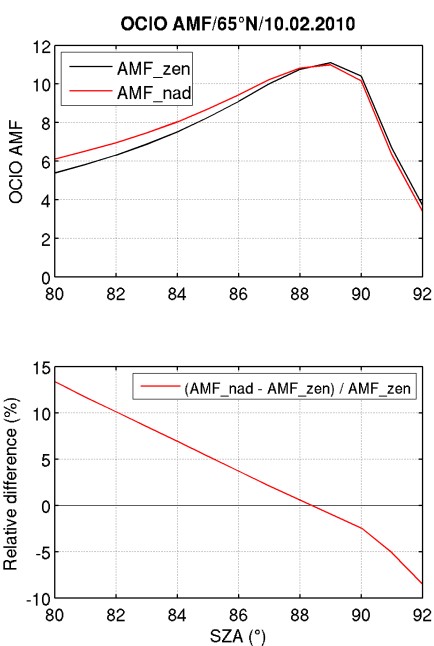

**Figure 8.** OClO AMF calculations for 65°N from ground-based zenith and satellite nadir geometries.

Zenith and nadir AMF calculations for one OClO activated day were performed here for conditions corresponding to 65°N, as shown in Figure 8. The simulations were performed using an implementation of the DISORT radiative transfer code accounting for the impact of photochemical enhancements along the light path at twilight (Hendrick et al., 2007). They confirm the Oetjen et al. (2011) results, with differences of up to 13% for SZA between 80 and 88°, and differences of up to -8% between 88.5 and 92° SZA. On average, over the 85° to 92° SZA range, the AMF difference is close to zero.





## 4   Comparison results

Figures 9 to 14 present respectively the time-series of the GOME-2A (2007-2016) and GOME-2B (2013-2016) together with ground-based OClO SCDs measurements performed in each hemisphere. As expected, the data from the four Antarctic stations (Fig. 9 and 10) show a stronger OClO signal in the winter months, with values up to 50-100 $\times 10^{13}$ molec/cm$^2$, when the stations are under the influence of the polar vortex.

The presence of larger OClO columns in the austral winter and spring compared to the Northern Hemisphere was highlighted in past satellite's studies (Wagner et al., 2001, 2002; Wittrock et al., 1999). Above the Antarctic, high OClO SCDs are usually observed after mid May, with a large increase within a few days, reaching a maximum by mid-September and then quickly decreasing until the chlorine activation stops by late October (Wagner et al., 2001; Richter et al., 2005). Due to a less stable polar vortex, the year-to-year variability of OClO is large in the Northern Hemisphere, so that only few years are characterised by large activation events (Richter et al., 2005). The yearly variability in OClO SCD is anti-correlated with the temperature variations and modulated by PSC formation (Weber et al., 2003).

### 4.1   Antarctic

Figure 9 presents the daily comparisons between GOME-2A and ground-based data at the four Antarctic stations. At the Neumayer station, the ground-based OClO SCDs are available for the complete period of GOME-2A observations (2007-2016) showing enhanced OClO signals between August and October, when the polar vortex is above the station. In Arrival Heights, comparisons during the activated periods are missing for two years (2008 and 2014) in the IUPH dataset but are covered by the NIWA measurements, while in Belgrano four years of data outside of the polar night period (mid April to end of August) are available for the GOME-2 comparisons and in Marambio only one year (2015). When ground-based data are not available, the satellite daily mean overpass are displayed in light red. Figure 10 similarly presents the time-series of the GOME-2B comparisons for 2013-2016.

Each year an enhanced OClO signal (from 20 and up to 40 and 60 $\times 10^{13}$ molec/cm$^2$) is observed in August and September, followed by a decrease. The largest OClO columns are measured at Arrival Heights in 2012, 2013, 2015, at Neumayer in 2013, 2014, 2015 and at Belgrano in 2011, 2014 and 2015. There is some variability in the strength of the signal from year to year, but the daily variations are sampled in a coherent way from the ground and from space, with a general tendency for smaller (sometimes negative, especially for GOME-2A) OClO SCDs retrieved by the satellites during November to April, outside of the chlorine activation period.

A gap in the GOME-2A data is observed in October at the Neumayer station since 2013, due to the reduced swath of the satellite instrument. There are no satellite measurements within 200 km for both sensors between May and end of July, which results in missing the start of the chlorine activation. Some more pronounced negative slant columns appear in the GOME-2A dataset after mid 2011, probably related to the degradation of the instrument. A quantitative comparisons for different GOME-2A periods is also shown in Fig. 15 and discussed later on.

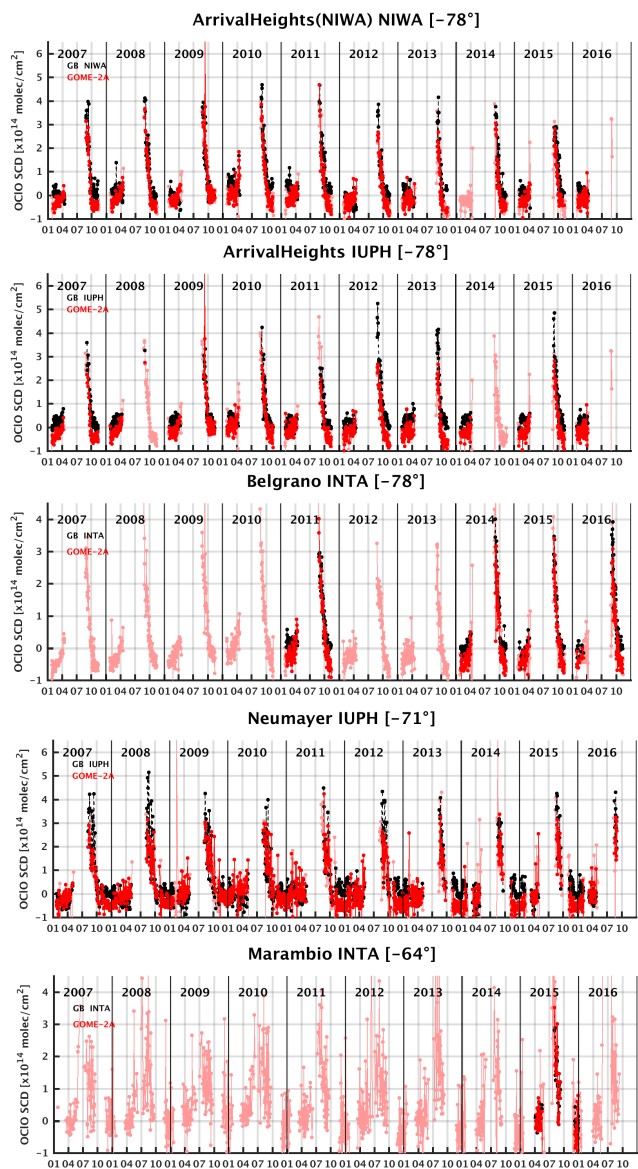

**Figure 9.** Time series of GOME-2A (red) OClO daily mean slant column data co-located with ground-based (black) measurements performed at each Antarctic station. Lighter/transparent red color is used for GOME-2A when there is no ground-based measurements.

In Marambio, an enhanced OClO signal is observed in June, August and September, with a data gap in July. A day-to-day variability of several $10 \times 10^{13}$ molec/cm$^2$ is visible in GOME-2B data (Figure 10) during the activated period. This behaviour is related to the intermittent probing of air masses that are on the edge of the Antarctic polar vortex. The ground-based data



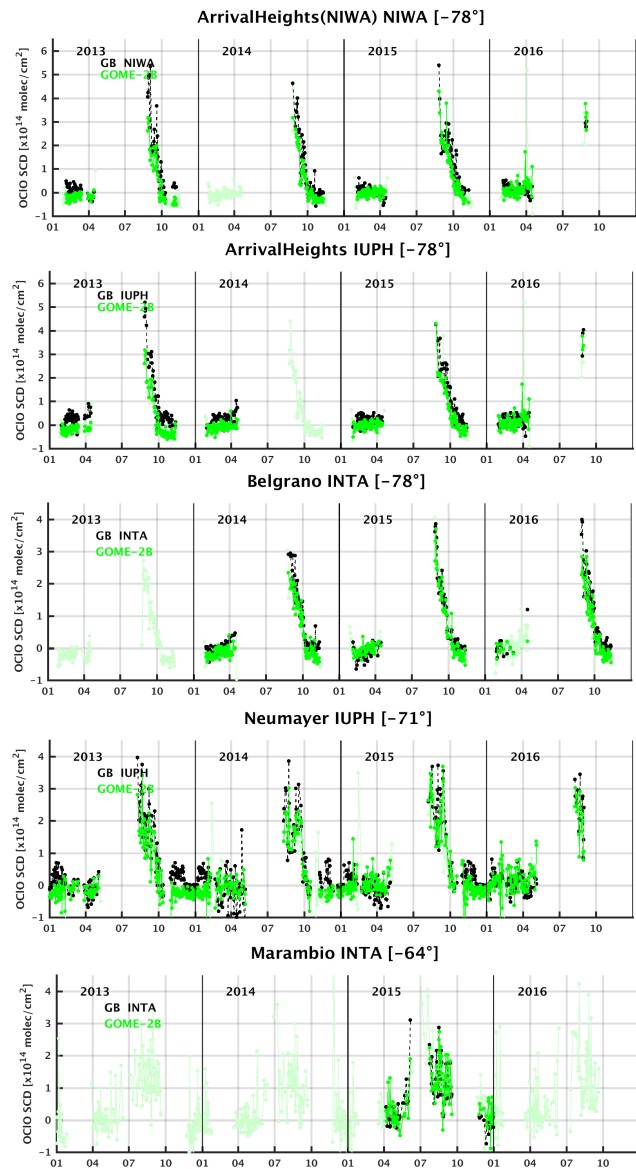

**Figure 10.** Time series of GOME-2B (green) OClO daily mean slant column data co-located with ground-based (black) measurements performed at each Antarctic station. Lighter/transparent green color is used for GOME-2B when there is no ground-based measurements.

seem more sensitive to these rapid changes, resulting in higher peaks than observed with GOME-2A and GOME-2B. For this station, the averaging of the satellite data within 200 km could mix air from inside and outside the vortex. Tests with a smaller co-location radius were performed for this station, but with similar results and less co-located points.





The statistical analysis (presented in Fig. 11 and Fig. 12) leads to correlation coefficients from 0.77 (Neumayer) to 0.92 (Belgrano) for GOME-2A and from 0.84 to 0.95 for GOME-2B daily comparisons, with linear regression slopes in the range

of 0.72-1.06 and 0.71-0.84 for GOME-2A and GOME-2B, respectively.

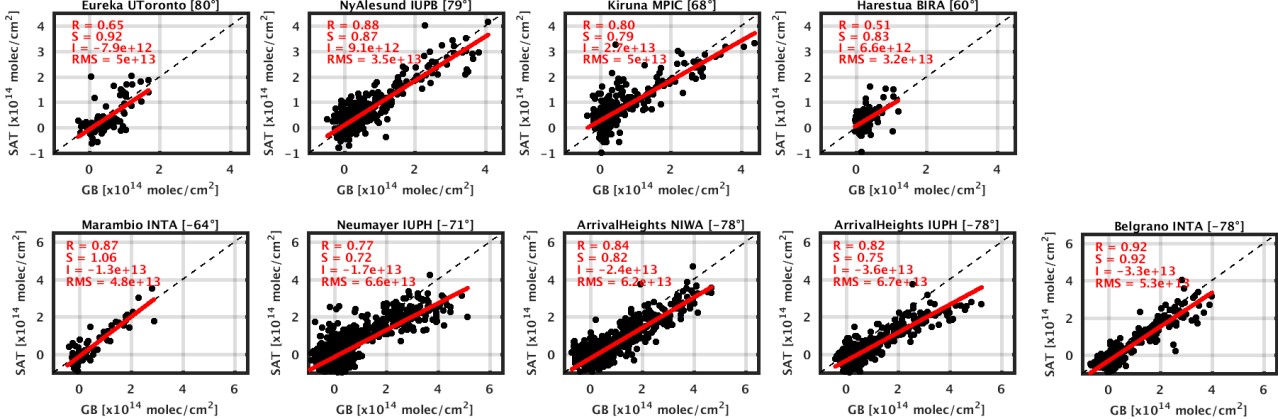

**Figure 11.** Scatter plot of GOME-2A OClO slant column data co-located with ground-based measurements at each station.

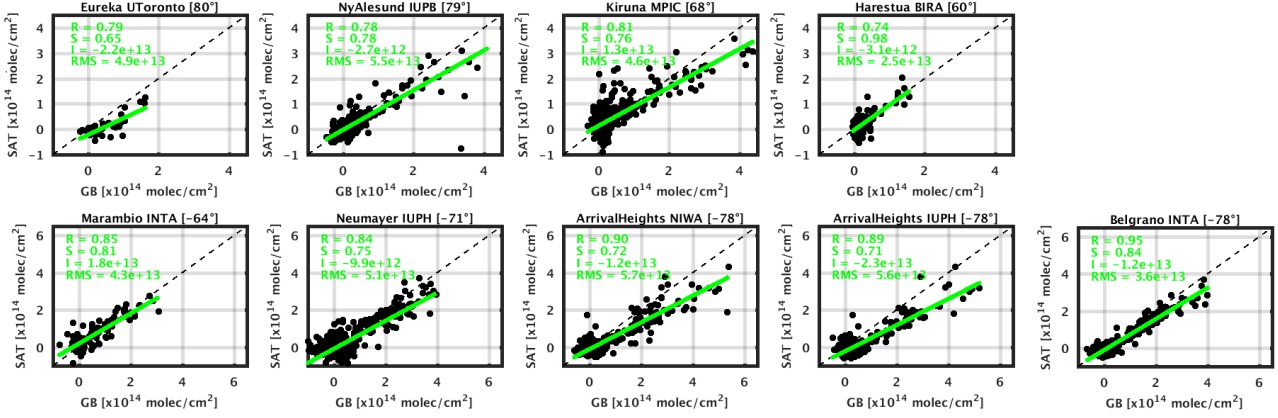

**Figure 12.** Scatter plot of GOME-2B OClO slant column data co-located with ground-based measurements at each station.



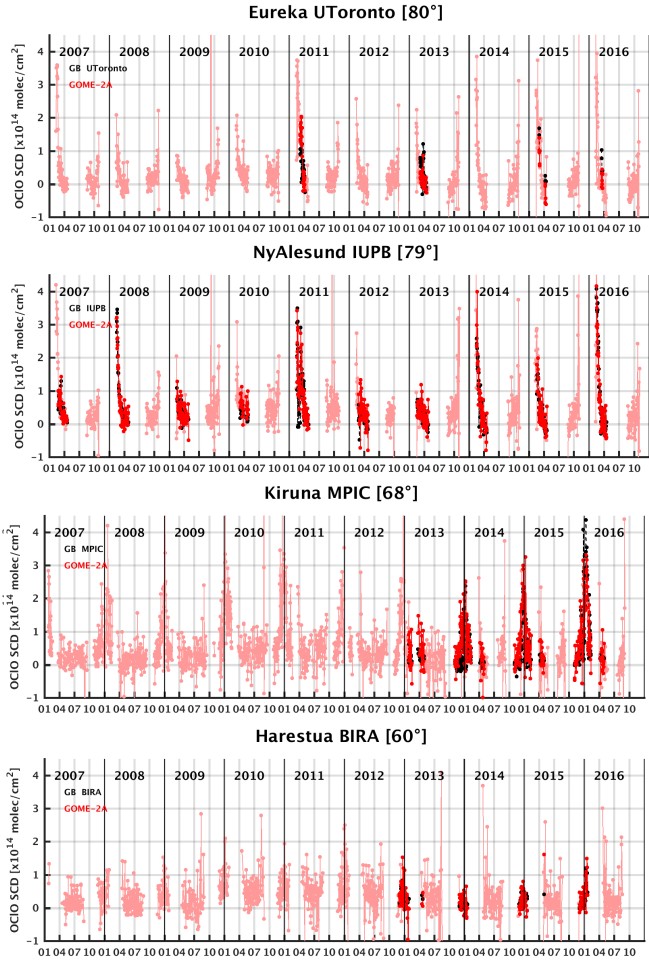

**Figure 13.** Time series of GOME-2A (red) OClO daily mean slant column data co-located with ground-based (black) measurements performed at each Arctic station. Lighter/transparent red color is used for GOME-2A when there is no ground-based measurements.

## 4.2 Arctic

Comparisons at the four Arctic stations are shown in Fig. 13 and 14. It should be noted that Eureka and Ny-Ålesund are in the polar night until about February/March, so that ground-based measurements can be done only up to April/May. After that period, SZAs are too low (smaller than 88°) to perform ground-based measurements of OClO.

At all stations GOME-2A, GOME-2B and the zenith-sky DOAS instruments capture similarly the seasonal cycle of the OClO SCD, as well as its day-to-day variations. Differences from year to year and station to station exist, but typically enhanced OClO slant columns are found at the four sites between October and March, with large values in 2007, 2008, 2011, 2014, 2015, 2016 and values up to 40 x$10^{13}$ molec/cm$^2$, as in Ny-Ålesund and Kiruna during the 2015-2016 winter.



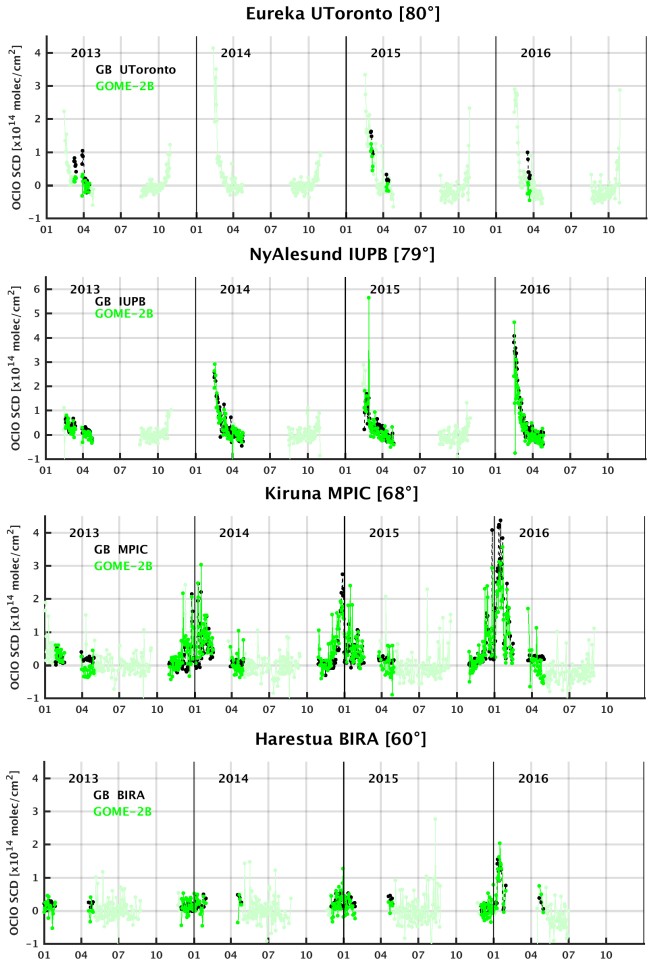

**Figure 14.** Time series of GOME-2B (green) OClO daily mean slant column data co-located with ground-based (black) measurements performed at each Arctic station. Lighter/transparent green color is used for GOME-2B when there is no ground-based measurements.

For Ny-Ålesund and Kiruna, years 2014, 2015 and 2016 show an enhanced OClO signal (with peaks larger than 20-30 $\times 10^{13}$

molec/cm$^2$) while 2013 does not seem to show any chlorine activation. Unlike Ny-Ålesund and Kiruna, the chlorine activation in 2014 and 2015 cannot be seen in Harestua, probably due to the lack of polar vortex excursions at latitudes as low as as 60°. In 2016, on the other hand, a clear enhancement is visible from the ground and from GOME-2A and GOME-2B in January (with a peak 13-15 $\times 10^{13}$ molec/cm$^2$). Unfortunately the gap in the GOME-2 data from February to May (related to the GOME-2 pixels SZA being smaller than 85°) prevents to detect the other OClO SCD peaks seen by the ground-based instrument.

The large OClO peak in early 2008 can be understood by the very cold stratospheric temperatures in the winter 2007/2008. According to Kuttippurath et al. (2009), the temperature started to decrease in November 2007 and remained low until a major





stratospheric warming in late February 2008. At this time, temperatures were below the PSC formation threshold inside the polar vortex. According to Tétard et al. (2009), in January 2008, the polar vortex was not centered on the geographical north pole, and was gradually moving towards Europe. This would bring the vortex over Ny-Ålesund and allow to measure high
OClO SCDs over this station.

GOME-2A is more noisy than GOME-2B, especially outside the chlorine activation period (e.g. negative points in January to April and after September in 2013 and in the following years), but both sensors follow nicely the enhanced OClO signals in winter periods. As for GOME-2A, the gap in the comparisons around February, March and part of April is related to the GOME-2B SZA being smaller than 85° in that period, leading to the exclusion of these data (see Sect. 2).

Differences between GOME-2A and GOME-2B are related to the smaller GOME-2A swath after July 2013 and the 30 minutes difference between both instrument's. Moreover, the GOME-2A degradation and the possible different impact of the mean residual, the scan angle empirical correction functions and the additional offset correction as discussed in Sect. 2 could also play a role in enhancing the noise of GOME-2A OClO columns in comparisons to GOME-2B.

An illustration of this time-degradation effect is given in Figure 15. On the left panel all the Neumayer data are presented
while, on the right panel, only the first 4 years are displayed for each instrument (2007-2011 and 2013-2016). RMS values given below the figures clearly demonstrate that during their first 4 years of operation, both GOME-2A and GOME-2B had a similar level of noise.

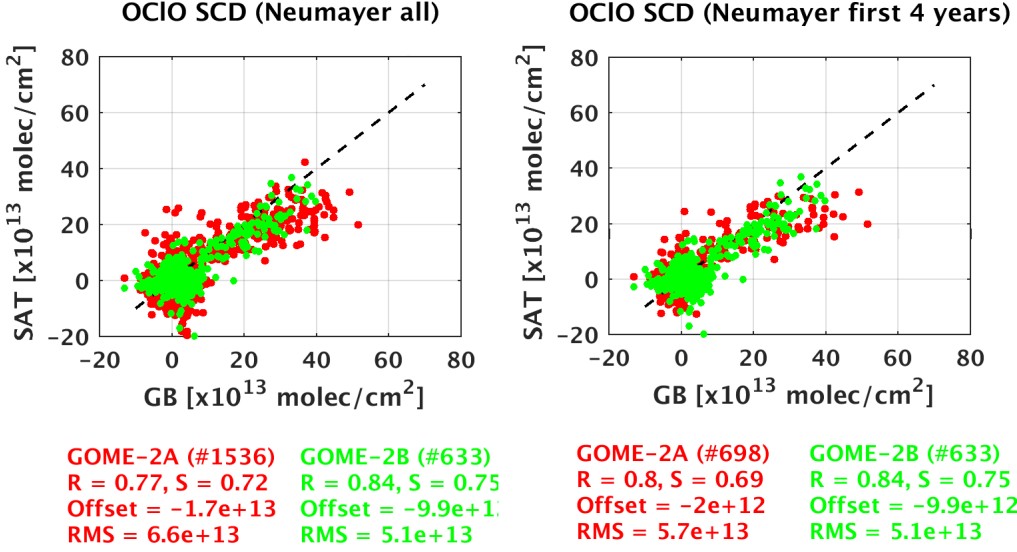

**Figure 15.** Scatter plot between daily GOME-2A (red) and GOME-2B (green) GDP 4.8 satellite data and ground-based data at Neumayer station for all data (left) and the first 4 years of operation (right) of each satellite.





The statistical analysis (presented in Fig. 11 and Fig. 12) leads to correlation coefficients from 0.51 (Harestua) to 0.88 (Ny-Ålesund) for GOME-2A and 0.74 to 0.81 for GOME-2B daily comparisons, with linear regression slopes around 0.79-0.92
and 0.65-0.98 for GOME-2A and GOME-2B respectively.

## 4.3 Comparison summary

We now consider all the stations and focus only on the activated periods (July-August-September in the Southern Hemisphere and January-February-March in the Northern Hemisphere). Figure 16 summarizes the biases between GOME-2 and ground-based ZSL-DOAS time-series using box-whisker plots of their differences at each site. Stations are ordered by latitude, from
the Arctic (top) to the Antarctic (bottom). It is worth mentioning that although Eureka and Ny-Ålesund are close to each other in latitude (80°N and 79°N), they are far away in longitude (Canada and northern Europe), which implies very different positions with respect to the polar vortex. This is also true for Arrival Heights and Belgrano, which are both at a latitude of 78°S but located at opposite sides of the Antarctic continent (see map in Figure 3). The figure indicates a general negative bias (up to around -8 x$10^{13}$ molec/cm$^2$) for both GOME-2 instruments at most stations, except for Kiruna and Marambio. The
differences between GOME-2A and GOME-2B are of a few $10^{13}$ molec/cm$^2$. Differences of the same order of magnitude are found e.g., between the two Arrival Heights instruments. The median bias statistics of the individual comparisons are reported in Table 4 for each station and for both hemispheres, together with regression analysis statistics. In relative values, the station biases range from -53% to 8% for GOME-2A and -78% to 13% for GOME-2B for Eureka and Marambio.

**Table 4.** Summary of the regression parameters and bias between GOME-2A and B and zenith-sky OClO SCDs daily mean comparisons for the active months (January-February-March for the Northerm Hemisphere and July-August-September for the Southern Hemisphere). Intercept, RMS and absolute biases (median (SAT-GB)) are in x$10^{13}$ molec/cm$^2$.

| Station | Period | GOME-2A Regression | | | | Bias | Period | GOME-2B Regression | | | | Bias |
|---|---|---|---|---|---|---|---|---|---|---|---|---|
| | | R | S | I | RMS | abs [rel] | | R | S | I | RMS | abs [rel] |
| Eureka | 2011-2016 | 0.55 | 0.83 | 0.19 | 5.5 | -2.8 [-53%] | 2013-2016 | 0.86 | 0.99 | -5.5 | 6.1 | -5.6 [-78%] |
| Ny-Ålesund | 2007-2016 | 0.89 | 0.86 | 1.2 | 3.8 | -0.05 [3.2%] | 2013-2016 | 0.74 | 0.78 | -0.14 | 6.7 | -1.9 [-33%] |
| Kiruna | 2013-2016 | 0.85 | 0.76 | 2.6 | 4.8 | 0.2 [0.09%] | 2013-2016 | 0.87 | 0.70 | 2.5 | 4.7 | -0.12 [-9%] |
| Harestua | 2012-2016 | 0.49 | 0.81 | -0.32 | 4.1 | -0.6 [-21%] | 2013-2016 | 0.87 | 1.04 | -0.61 | 2.26 | -0.5 [-21.5%] |
| Marambio | 2015 | 0.86 | 1.01 | -0.1 | 4.6 | 0.9 [8%] | 2015 | 0.88 | 0.89 | 1.8 | 3.44 | 1.1 [13%] |
| Neumayer | 2007-2016 | 0.75 | 0.51 | 5.9 | 8.6 | -3.5 [-18%] | 2013-2016 | 0.85 | 0.76 | 0.82 | 6.2 | -3.5 [-20%] |
| ArrivalHeights (NIWA) | 2007-2016 | 0.74 | 0.77 | 0.29 | 8.9 | -6.9 [-36%] | 2013-2016 | 0.84 | 0.71 | -0.24 | 9.1 | -6 [-37%] |
| ArrivalHeights (IUPH) | 2007-2016 | 0.65 | 0.64 | -0.09 | 11.7 | -8.3 [-42%] | 2013-2016 | 0.89 | 0.74 | -2.5 | 10.3 | -7.5 [-40%] |
| Belgrano | 2011;2014-2016 | 0.77 | 0.76 | 0.68 | 6.9 | -3.5 [-23%] | 2014-2016 | 0.91 | 0.76 | 1.03 | 4.96 | -3.3 [-19%] |
| all stations/points | 2007-2016 | 0.80 | 0.64 | 2 | 7.9 | -2.3 [-22.3%] | 2013-2016 | 0.87 | 0.73 | 0.9 | 6.3 | -2.2 [-24.4%] |
| NH stations/points | 2007-2016 | 0.85 | 0.85 | 1.1 | 4.3 | -0.1 [-5%] | 2013-2016 | 0.79 | 0.76 | 0.64 | 5.3 | -1.1 [-24%] |
| SH stations/points | 2007-2016 | 0.71 | 0.61 | 2.5 | 9.9 | -5.5 [-30%] | 2013-2016 | 0.84 | 0.70 | 1.3 | 7.25 | -4.4 [-24%] |



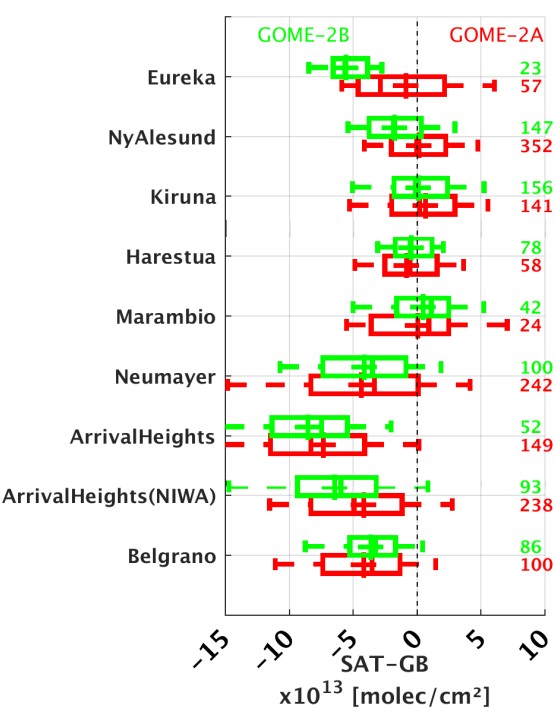

**Figure 16.** Box and whisker plot of the difference between all the GOME-2 and ZSL-DOAS OClO SCD pairs during active period months. Stations are ordered by decreasing latitude (South at the bottom). The box and whisker plots are defined as follow: crosses and lines for the mean and median values, boxes for the 25th and 75th percentile and dashed lines for the 9th and 91st percentile. Numbers on the right correspond to the number of days considered in the analysis.

Figure 17 presents the results as a scatter plot, with GOME-2A values in red and GOME-2B values in green. It can be
seen that GOME-2A results are slightly noisier than GOME-2B, with several outliers, a smaller correlation coefficient (0.8 wrt to 0.87) and larger RMS values. As already mentioned, this is likely related to instrumental degradation effects and/or the different empirical corrections used for GOME-2A. Regression slopes are about 0.64 for GOME-2A and 0.72 for GOME-2B, with an intercept of about $2 \times 10^{13}$ molec/cm$^2$ for GOME-2A and half of it for GOME-2B. Fig. 18 presents the same data but color-coded according to the different stations.

Concentrating on the slopes of daily linear regressions at each station (Table 4), values around or better than 0.7 are found for GOME-2B, and often slightly smaller for GOME-2A. The intercepts are generally smaller than $2 \times 10^{13}$ molec/cm$^2$, except at Kiruna (for both instruments) and at Neumayer for GOME-2A. RMS are generally larger for Antarctic stations.





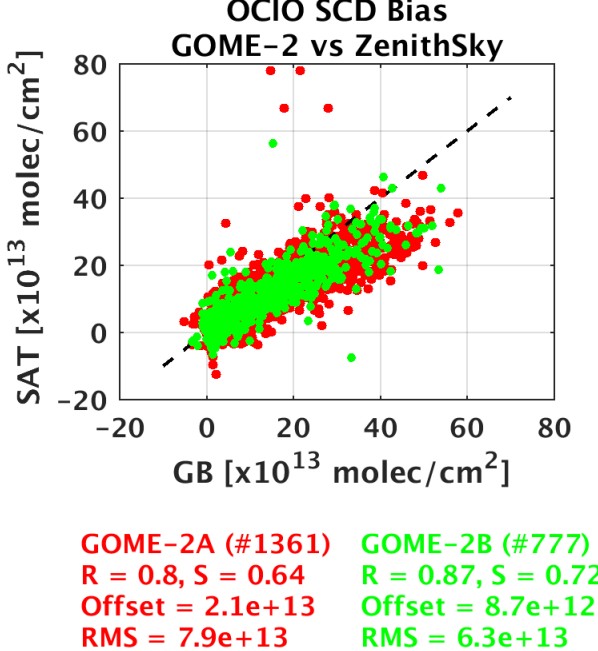

**Figure 17.** Scatter plot between daily GOME-2A (red) and GOME-2B (green) GDP 4.8 satellite data and ground-based data for all the stations included in the study, during active period months.

These results are to be put in perspective with the systematic bias estimated in Sect. 3.1.1 and summarized in Table 3. Some stations have larger expected biases than others (e.g. Eureka up to 15%) due to their DOAS settings choices, and in general,

there is a total uncertainty within the ground-based datasets of about 26 to 33%.

When considering results grouped by hemisphere, the slope is larger in the northern hemisphere for GOME-2A (0.85 wrt 0.61), while for GOME-2B results are more coherent (0.76 and 0.7). For GOME-2B the relative bias is very similar in both hemispheres (around -24%), while for GOME-2A it is about -5% in the northern hemisphere and -30% in the southern hemisphere.

To summarize, we can conclude that:

- The variability of the OClO column, from day-to-day fluctuations to the annual cycle, is captured consistently by all instruments.

- GOME-2A tends to be noisier than GOME-2B after late 2011.





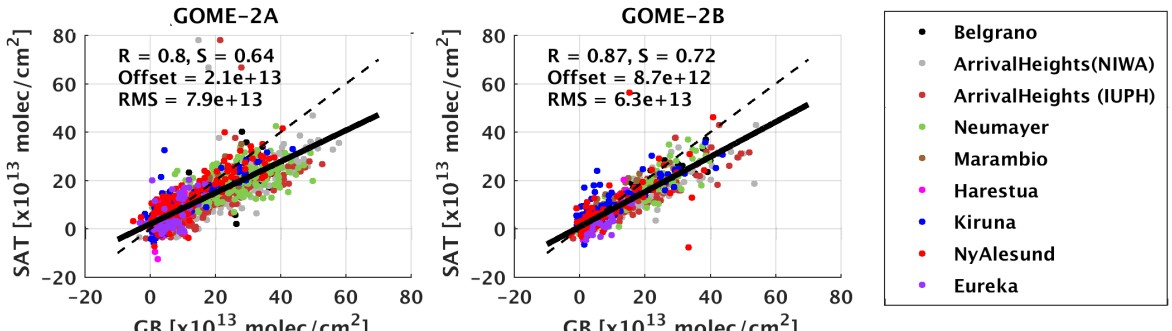

**Figure 18.** Scatter plot between daily GOME-2A (left) and GOME-2B (right) GDP 4.8 satellite data and ground-based data at the different stations included in the study, for the activated months (JAS for stations in the SH and JFM for stations in the NH). The stations are color-coded, and the total regression statistics are given as insert.

## 5   Conclusions

We investigated the quality of the GOME-2A (2007-2016) and GOME-2B (2012-2016) OClO GDP 4.8 slant column datasets by comparing them to ground-based ZSL-DOAS measurements at a selection of 8 stations located in the Arctic and Antarctic regions: Eureka (80°N), Ny-Ålesund (79°N), Kiruna (68°N), Harestua (60°N), Marambio (64°S), Neumayer (71°S), Belgrano (78°S) and Arrival Heights (78°S).

For ground-based instruments, OClO spectral analyses were performed using fixed noon spectra recorded at low SZA in
the absence of chlorine activation. Different DOAS analysis settings are used by different instrument teams, and the impact of these differences are quantified through dedicated sensitivity tests. This leads to an estimation of systematic uncertainties of about 25% maximum. Depending on the different instruments, the random noise error was estimated to be between 6 and 22%. The total uncertainty from each ground-based dataset is estimated to be between 26 to 33%, depending on the site.

At each station, daily comparisons were performed by selecting satellite and ground-based SCD data pairs corresponding to
similar SZA conditions, assuming similar AMFs in both nadir and zenith geometries. Using radiative transfer simulations, this assumption was shown to be valid within the SZA range of the measurements, confirming estimations from previous studies.

Daily mean OClO SCD time-series show that satellite and ground-based observations agree well at all stations, and display consistent seasonal and inter-annual variabilities. GOME-2A tends to be noisier than GOME-2B especially after 2011, which is likely related to instrumental degradation effects combined with the possible impact of the different instrumental corrections
applied to the two instruments.

Daily scatterplots based on data selected during chlorine activated periods give correlation coefficients of 0.8 for GOME-2A and 0.87 for GOME-2B, and regression slopes are 0.64 for GOME-2A and 0.72 for GOME-2B. These results fulfill the GOME-2 accuracy requirements for OClO, as stated in the EUMETSAT AC SAF Product Requirement Document, i.e. a target accuracy of 50% and an optimal accuracy of 30%.





Biases at each station are generally negative and close to -8 $\times 10^{13}$ molec/cm$^2$ in the worst case (Arrival Heights IUPH). Those biases do not seem to originate from the ground-based datasets since these were also used recently for TROPOMI OClO validation (Pinardi et al., 2020), showing excellent agreement. Overall, comparison points at all the stations display a median bias of about -2.2 $\times 10^{13}$ molec/cm$^2$ for both GOME-2 instruments.

    We conclude that the AC SAF 2007-2016 GOME-2 GDP 4.8 OClO SCD data records (publicly available through the
acsaf.eoc.dlr.de FTP server) meet AC SAF mission requirements for both OClO GOME2 product, but show an under-estimation of about 20-25% with respect to reference ground-based data.

    Room exists for further improvement of both satellite and ground-based data sets. An harmonization of ground-based zenith-sky analysis, e.g., by NDACC would be desirable when possible, considering the different spectral ranges covered by the different instruments. Moreover, 3D Chemistry Transport Model output coupled to a suitable radiative transport model could
allow creating meaningful OClO AMFs to transform the SCD OClO product into a more directly exploitable VCD product.

**Appendix A: Ground-based**

**A1   Ground-based sites description**

For this study, stations operating Zenith-Scattered Light (ZSL)-DOAS UV-Visible spectrometers from the Network for the Detection of Atmospheric Composition Change (NDACC, https://www.ndaccdemo.org/, last access on 28 June 2021), situated
above 60° latitude (North and South) and performing OClO SCD data retrievals have been selected.

    In the Arctic:

- UToronto operates the PEARL UV-VIS spectrometer at Eureka (80°N, 85.93°W, Nunavut, northern Canada). OClO SCD data have been analysed since 2011.

- IUP-Bremen operates a UV-VIS spectrometer at Ny-Ålesund (78.9°N, 11.9°E, Spitsbergen) since 1995 (Wittrock et al., 2004; Tørnkvist et al., 2002). OClO SCDs have been analysed since 2007 using one fixed reference for each season.

- MPIC operates a UV-VIS spectrometer at Kiruna (67.8°N, 20.4°E, Sweden) since 1996 (Gu, 2019; Bugarski, 2003; Gottschalk, 2013). OClO SCDs have been analysed since 2007, but between 2007 and 2013 the instrument was not operated on many days due to detector problems that prevented the OClO analysis.

- BIRA-IASB operates a UV-VIS spectrometer at Harestua (60.22°N, 10.75°E, Norway) since the nineties (Hendrick et al., 2007). End of 2012 a new instrument has been installed with an improved signal to noise ratio, and OClO SCDs have been analysed since then using annual reference spectra.

    In the Antarctic:

- IUP-Heidelberg operates a UV-VIS spectrometer at the German Antarctic research station Neumayer (70.62°S, 8.27°W,
on the ice shelf in the Atlantic sector of the Antarctic continent) since the 1999 (Frieß et al., 2004, 2005). OClO SCDs





have been analysed since 2007 using several fixed reference spectra. Generally enhanced OClO signals are observed between August and October, when the polar vortex is over the station.

- IUP-Heidelberg and NIWA jointly operate a UV-VIS spectrometer at Arrival Heights (77.83°S, 166.65°W), part of the New Zealand station Scott Base on Ross Island since 1998 (Frieß et al., 2005). Another instrument was present at the station, operated by NIWA (Kreher et al., 1996), but stopped measurements in 2017. Both instruments provide OClO SCDs since 2007.

- In 1995, INTA installed a zenith-DOAS Vis at Belgrano II station (77.9°S, 34.6°W), the Argentinian station situated on the coast of the Antarctic continent in the Weddell Sea area (Yela et al., 2005, 2017). Belgrano is representative of an in-polar vortex station during winter-spring season until the vortex breakdown (Yela et al., 2005, 2017). In 2011, a MAX-DOAS UV/Vis was installed at Belgrano II (Prados-Roman et al., 2018; Gomez-Martin et al., 2021). OClO SCD have been analysed in the UV channel for 2011, 2014, 2015, 2016, 2018 and 2019. Ground-based SCDs measurements are made for SZA<92º, with no measurements during the polar night period (mid-April to mid-August).

- In 1994, INTA installed a zenith-DOAS Vis at Marambio station (64.3°S, 56.7°W), in Marambio Island (Yela et al., 2017). In 2015, a MAX-DOAS UV/Vis was installed in the same site (Prados-Roman et al., 2018). Marambio is frequently located in the vortex edge region and affected by both vortex air masses and mid-latitude air masses (Aun et al., 2020). OClO SCD have been analysed in the UV channel for 2015 and for 2018 onward.

**Table A1.** Information on ground-based DOAS instruments.

| Station | Group | Coordinates | Resolution [nm] | wvl range [nm] |
|---|---|---|---|---|
| Eureka | UToronto | 80.05°N, 86.42°W | 0.5 | 320-400 |
| Ny-Ålesund | IUPB | 78.9°N, 11.9°E | 0.5 | 302-390.8 |
| Kiruna | MPIC | 67.8°N, 20.4°E | 0.6 | 300-400 |
| Harestua | BIRA | 60.2°N, 10.7°E | 0.5 | 290.2-379 |
| Belgrano | INTA | 77.9°S, 34.6°W | 0.5 | 320.5-415.5 |
| Marambio | INTA | 64.3°S, 56.7°W | 0.5 | 327.5-407.5 |
| Neumayer | IUPH | 70.6°S, 8.3°W | 0.5 | 320-420 |
| Arrival Heights | IUPH | 77.8°S, 166.6°W | 0.5 | 320-420 |
| Arrival Heights | NIWA | 77.8°S, 166.6°W | 0.56 | 402-440 |

## A2  Sensitivity tests

*Systematic errors:*





In a first test, OClO SCD analysis are performed in the 345-389nm range (as for the GOME-2 analysis window, with varying
OClO cross-section sources (using the Wahner et al. (1987), the Kromminga et al. (1999) and the Kromminga et al. (2003)
cross-sections at several temperatures), and fixing the other inputs, as summarized in Table A2. With respect to Kromminga
et al. (2003) at 213K (used for GOME-2 analysis), regression analysis reveals slopes of 1.02 for the Kromminga et al. (2003)
at 233K, 0.97 for the Kromminga et al. (1999) also at 213K and of 0.85 for the Wahner et al. (1987) at 204K (see Fig. A1),
so a total uncertainty of about 15% with respect to what used for GOME-2 retrievals. This is coherent with Kromminga et al.
(2003) reporting cross-section band peaks about 8% smaller than Wahner et al. (1987).

Considering the largest impact between results obtained with the different OClO cross-sections, we come to a difference
of about 17% (corresponding to slopes ranging from 0.85 to 1.02). This value is used to quantify the first component of the
systematic uncertainty in Table A2. The expected bias for each group OClO cross-section choice is also reported for each
station in Table 3.

For the second test (see Table A2), we fixed the OClO cross-section to Kromminga et al. (2003) at 213K and varied the other
DOAS fit parameters in an attempt to match the different settings used by each group (wavelength interval, interfering species
and their cross-section references as in Table 2). Unfortunately, the Ny-Ålesund instrument does not cover the visible range
and stops at 390.4nm and the MPIC wavelength choice (interval 372-392nm) cannot be entirely covered. It should be noted
that no analysis could be done in the visible interval used by NIWA.

Results of the regression analysis for each group choice with respect to the median OClO SCD values, are presented in
Figure A2. In most cases, the regression is compact (correlation R larger than 0.945) except for MPIC (R=0.893), also the RMS
is generally smaller than $2x10^{13}$ molec/cm$^2$, except for IUPH and MPIC. Results for the latter two cases are likely biased due
to the limited wavelength range (up to about 390.4nm) of the Ny-lesund spectra. As a result in these cases, the upper part of the
wavelength interval is not covered. Depending on the setting choices, the difference compared to the median OClO SCD can
take the form of a multiplicative bias (slope different than 1) and/or an additive bias (non-zero intercept). In the tested cases,
all intercept except for IUPH are smaller than $1x10^{13}$ molec/cm$^2$, so the observed bias is mostly multiplicative. The largest
impact on the slope is obtained for the MPIC case (slope of 0.925) and for UToronto (1.04), leading to a difference between
all cases of about 18.5% (slopes from 0.925 to 1.11). This value is considered as the maximum systematic uncertainty on the
retrieval choice for the systematic uncertainty contribution in Table A2, leading to a total maximum systematic uncertainty of
about 25% (see Table A2).

*Expected systematic bias against GOME-2:*

A third test has been carried out (see Table A2), comparing each group analysis to the OClO SCD obtained using the GOME-
2 data retrieval settings (345-389nm range, see Table 1), as illustrated in Fig. A3. From this sensitivity test, the expected
systematic bias for each group is estimated in comparison to the GOME-2 retrieval settings, ranging between 4% and 16% for
the different stations.





**Table A2.** Description of the different sensitivity tests performed and main results summary. The letters refer to Table 2.

| Tests | wavelength range (nm) | Cross-sections | | | | | | slopes min, max | Syst. Contr. |
|---|---|---|---|---|---|---|---|---|---|
| | | OClO | NO$_2$ | O$_3$ | BrO | O$_4$ | Ring | | |
| 1) OClO cross sections | 345-389 | [a] (204K), [b] (213K), [c] (213K,233K) | (220K)* Gur2005 | (223K, 243K)* Gur2005 | - | [l] (298K) | Ring Vountas | 0.85, 1.02 | 17% |
| 2) retrieval choices vs median | as Table 2 | [c] (213K) | | as Table 2 | | | | 0.925, 1.11 | 18.5% |
| | | | | | | | | | 25 % |
| 3) retrieval choices vs GOME-2 (Table1) | as Table 2 | [c] (213K) | | as Table 2 | | | | 0.84, 1.03 | 19% |

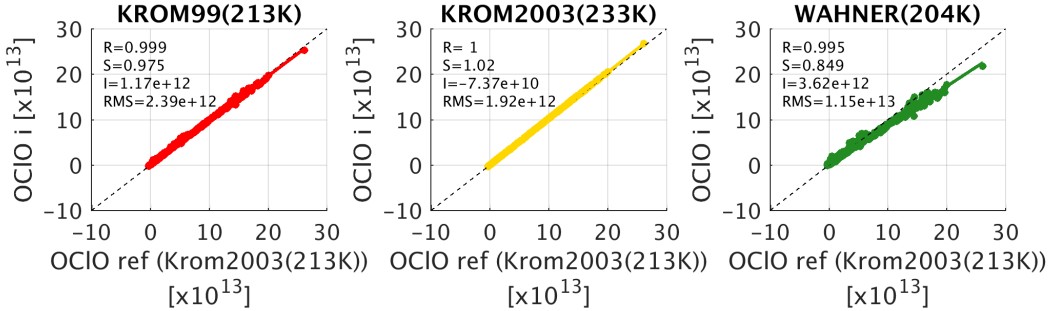

**Figure A1.** Regression analysis of OClO SCD retrieved from a common set of Ny-lesund spectra to investigate the sensitivity of OClO results on cross-sections used. The different DOAS analysis used correspond to what described in Table A2, for tests 1), with respect to OClO values obtained using the Kromminga et al (2003) cross-section at 213K as in GOME-2.

*Author contributions.* GP carried out the validation analysis, the associated investigations and wrote the manuscript. MVR and FH contributed input and advise at all stages of the scientific discussions and of the manuscript writing. MVR prepared the ground-based offset correction, FH performed the AMF calculations and JG pre-processed the satellite data. AR and PV developed the GOME-2 OClO data processor. MVR, FH, AR, FW, UF, RQ, PJ, KB, KS, MG, TW, MYG, CPR analysed the ground-based data and/or supervised the instrument operation. All co-authors revised and commented on the manuscript.

*Competing interests.* The authors declare that they have no conflict of interest.




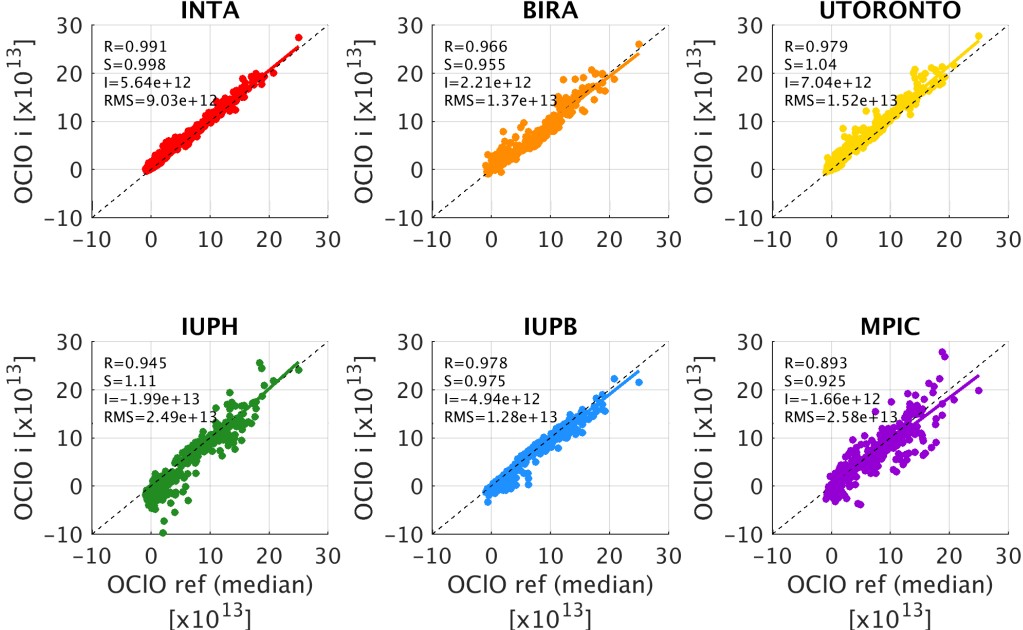

**Figure A2.** Regression analysis of OClO SCD retrieved from a common set of Ny-lesund spectra to investigate the sensitivity of OClO results on different settings. The different DOAS analyses used correspond to those used by each group for their own station analysis, as described in Table 2 and A2 in tests 2). Each set of OClO SCD is compared against median OClO values and regression statistics are given as inset in each plot.

*Acknowledgements.* Part of the reported work was carried out in the framework of the EUMETSAT AC SAF Continuous Development and Operations Phase (CDOP-2 and -3), and by the Belgian Federal Science Policy Office (BELSPO) via the ProDEx BeACSAF contribution to
the AC-SAF. EUMETSAT and the AC SAF are acknowledged for the production of GOME-2 GDP 4.8 data.

The authors are also grateful to O. Rasson for valuable IT support and for their dedication to the AC SAF operational validation.

The ZSL-DOAS data used in this publication were obtained from the PIs, and stations are part of the Network for the Detection of Atmospheric Composition Change (NDACC, https://ndacc.org). The ZSL-DOAS instrument PIs and staff at the stations are warmly thanked for their sustained effort on maintaining high quality measurements and for valuable scientific discussions. MPIC whish to thank Carl-Fredrik
Enell and Uwe Raffalksi for operating the Kiruna DOAS instrument. The ZSL-DOAS measurements at Eureka were made at the Polar Environment Atmospheric Research Laboratory (PEARL) by the Canadian Network for the Detection of Atmospheric Change (CANDAC), primarily supported by the Canadian Space Agency, the Natural Sciences and Engineering Research Council, and Environment and Climate Change Canada. INTA's observations were funded by the Spanish Ministry of Science and Innovation under the projects VHODCA (CTM2017-83199P), HELADO (CTM2013-41311P) and VIOLIN (CGL2010-20353). NIWA measurements at Arrival Heights are supported
through New Zealand's Ministry of Business, Innovation and Employment (MBIE) Strategic Science Investment Fund (SSIF).



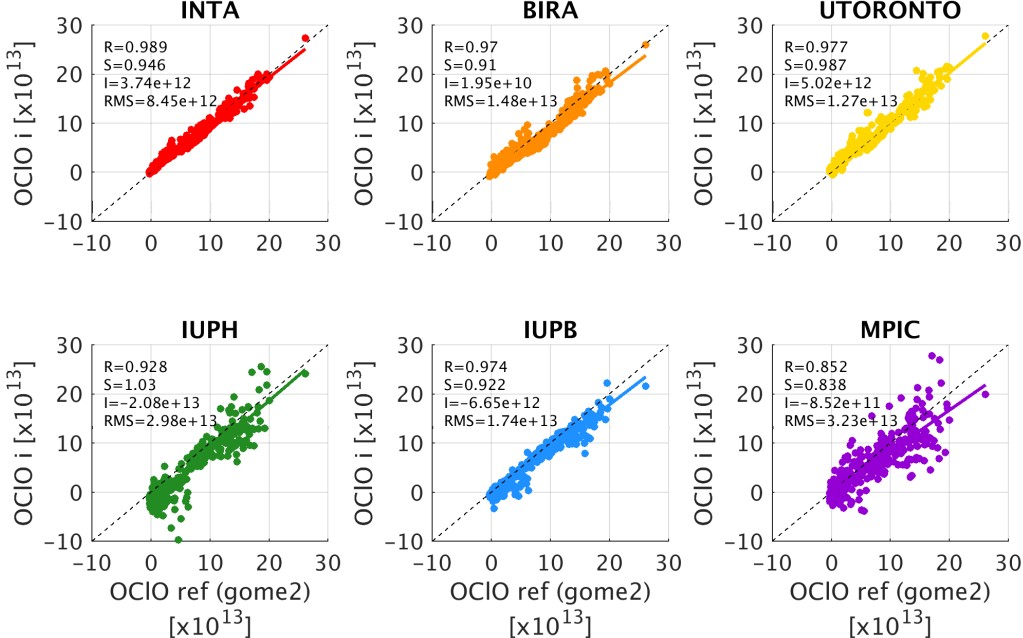

**Figure A3.** Regression analysis of OClO SCD retrieved from a common set of Ny-lesund spectra to investigate the sensitivity of OClO results on different settings. The different DOAS analysis used correspond to what each group used for their own station analysis, as described in Table 2 and A2 in test 3). Each set of OClO SCD is compared against the OClO values obtained using the GOME-2 retrieval settings described in Table 1 and regression statistics are given as inset for each plot.

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
