# Peer review of "Ground-based validation of the MetopA and B GOME-2 OCIO measurements"

_Atmospheric Measurement Techniques, 2021_

## Referee Comment (RC1)

This manuscript presents a validation exercise for the GOME2-A and GOME2-B OClO data product using OClO SCDs measured at 9 high latitude NDACC stations. Given the range of parameters used in the different data analysis approaches undertaken by the individual research groups for each of the stations, the sensitivity tests performed as part of this study are essential for a meaningful outcome. The authors found that the total uncertainty for the OClO data sets investigated in this study ranges from about 26% to 33% for the different stations. They furthermore found that satellite and ground-based data sets show a good agreement for the inter-annual variability and the overall seasonal behaviour at the different sites. But they also found a median bias of about -2.2x10$^{13}$ molec/cm$^2$ over all stations for both GOME-2 instruments with individual biases up to 8x10$^{13}$ molec/cm$^2$.

The validation study is comprehensible and clearly presented in the manuscript, and the authors also provide a more in-depth description of the three sensitivity tests in Appendix A2. The paper is definitely recommended for publication in AMT after the specific comments below have been addressed.

Specific comments:

Page 1, line 1: '… produced within the …'

Page 1, line 3: Only measurements up to 2016 are discussed in this paper. Why was this study not extended to include at least some data from the most recent 5 years (2017 – 2021)?

Page 1, lines 6-13: The uncertainty for the ground-based data sets is provided in the abstract as a percentage (lines 6/7) while the bias between ground-based and satellite data is given as an absolute number (lines 11/13). It would certainly be helpful if one of the two quantities could be provided as both, percentage and absolute value. That would make it easier to understand and interpret the information provided in the abstract, and it would put the retrieved bias and the uncertainty into context.

Page 1, line 7: '… data analyses …'

Page 2, line 19: '…associated with strong …'

Page 2, line 35: '…its Amendments.'

Page 3, line 51: delete 'study'

Page 3, line57: '… mostly for a few …

Page 3, line 59: 'In this paper, …'

Page 3, line 60: Add space between 'AC' and 'SAF'

Page 3, line 63: '… comparison method.'

Page 4, line 92: Replace comma with space after 'orbit'

Page 4, lines 90-93: Would be great, if you could give the reader an idea here regarding how big the amount in this bias correction is compared to actual OClO amount? E.g. how does this amount compare with the median bias quoted in the abstract.

Page 5, Figure 1: It would really help with the readability of the plot if the text and legend would be bigger. Also add 'SCD' after 'OClO' in the caption.

Page 5, line 96: Add comma after 'circumstances'

Page 5, lines 103 & 105 & 107: Capitalize 'Hemisphere' when its used in combination with 'Southern' or 'Northern'.

Page 5, lines 106 – 108: Not sure if I quite follow this interpretation here. For me, it looks more like GOME-2A for the NH starts with a baseline close to 0 for the first 3 years, then has a jump up in 2010 before it slowly drifts down again to a 0 baseline in 2016. GOME2-A for the SH starts negative, drifts up until in it is in the positive in 2010/2011, but then jumps straight down again in 2011/12 and stays in the negative.

Page 6, line 127: comma after 'From Table 2'

Page 7, Figure 4 & Fig 4 caption: I like Figure 4, it's a nice visualisation of the different wavelength intervals used. To figure out which interval is used by which group, this can be identified via Table 2. Just to make it a bit easier, would it be possible to add the group names into Fig 4 straight behind the wavelength interval? Or alternatively, the group names could also be added in the caption e.g. in the order of appearance from top to bottom.

Page 7, Fig 4 caption: add 'analysis' after 'DOAS', just to clarify that this is not the wavelength interval each instrument covers but the interval each group uses for their data analysis.

Page 7, line 138: 'Also, …'

Page 8, line 139: Add comma after 'needed'

Page 8, line 141: Just to be clear, water vapour should have been included but it was not, correct? Could clarify that in the text.

Page 8, line 146: 'In this section, …'

Page 9, line 160-161: Is there any particular reason why Ny- Ålesund was chosen to be the test site?

Page 9, lines 168: How were the median OClO SCD values determined, e.g. were any selection criteria applied?

Page 10, line 192: '… lead to a systematic …'

Page 10, line 205: '… used as input for…'

Page 11, line 218: '… measurements at Arrival Heights. At this site, the …'

Page 12, Figure 7 caption: '… the offset …' and delete 'of' before 'Neumayer'

Page 13, line 238: delete the 2nd 'et al.'

Page 13, line 242 – 244: The authors state: 'On average, over the 85o to 92 o SZA range, the AMF difference is close to zero.' However, looking at Figure 8, this is still between 5% and -8% … is that accounted for?

Page 14, line 247: '… OClO SCD measurements…'

Page 14, line 252: '... mid-May…'

Page 14, line 254: '… is larger in …'

Page 14, line 255: '…OClO SCDs…'

Page 14, line 260: 'At Arrival Heights, …'

Page 14, line 262: '… mid-April…'

Page 14, line 264: '…overpasses are …'

Page 14, line 266: 'Each year, …'

Page 15, Fig 9 & page 16, Fig 10 captions: '…there are no …'  Same also for Figures 13 & 14

Page 18, line 288: '… can only be made during April/May …'

Page 18, line 291: '… SCDs …'

Page 18, line 299: '… prevent detection of the other …'

Page 19, line 300: 'The large OClO peak at Ny- Ålesund and Kiruna in early 2008 …' (just to be clear)

Page 20, line 304: Should it be 'over Ny- Ålesund and Kiruna' rather ?

Page 20, line 309:  Shouldn't that be 'GOME2-A SZA'?

Page 20, line 315: Add 'respectively' in the bracket

Page 20, Figure 15: Would be interesting to have the same plot for one more station, in particular e.g. for Ny- Ålesund (NH station).

Page 22, Fig 16 caption: '… defined as follows:…'

Page 23, Fig 17 caption: '… during the active months.'

Page 24, line 359: 'For the ground-based …'

Page 25, line 377: Replace 'points' with something like 'measurements' or 'data'

Page 25, line 380: 'OClO GOME2 products …'

Page 25, line 379-381: On what study or analysis is the conclusion based that the GOME2 OClO data product discussed within this manuscript meets the AC SAF mission requirements? Either this needs to be explained in more detail in the text or the relevant reference together with a short summary needs to be provided.

Page 25, line 390: comma after 'retrievals'

Page 24, line 401: 'At the end of 2012, a new instrument was installed …'

Page 24, line 405: '… since 1999 ….' (delete 'the')

Page 26, line 406: 'Generally, …'

Page 26, lines 412 & 418: '… a Vis zenith-sky DOAS at …' (more consistent with the rest of the text)

Page 26, line 414: '… during the winter/spring season …'

Page 26, lines 415 & 419: '… UV/Vis MAX-DOAS was…'

Page 26, lines 415 – 417: The last 2 sentences in this paragraph should be switched around.

Page 26, lines 415 & 421: 'OClO SCDs'

Page 26, line 416: 'Ground-based SCD measurements…' (delete s in SCDs)

Page 27, line 424: '… OClO SCD analyses …' and looks like a bracket is missing after 'window'

Page 27, line 429: '… what is used for …'

Page 27, line 433: '… each group's OClO cross-section …'

Page 27, line 440: '… each group's choice …'

Page 27, line 443: '… the Ny-Ålesund …'

Page 27, line 452: '… each group's analysis …'

Page 28, Figure A1, caption: '… Ny-Ålesund … result on the cross-sections … DOAS analyses …. what is described in …'

Page 29, Figure A2 & Figure A3, caption: '… OClO SCDs …' and '… Ny-Ålesund …'

Page 29, Figure A3, caption: '… DOAS analyses used ….'

---

## Author Comment (AC1)

Answer to RC1:

Reviewer comments are given in black and author answers are in blue. Changes in the revised manuscript are marked in red.

This manuscript presents a validation exercise for the GOME2-A and GOME2-B OClO data product using OClO SCDs measured at 9 high latitude NDACC stations. Given the range of parameters used in the different data analysis approaches undertaken by the individual research groups for each of the stations, the sensitivity tests performed as part of this study are essential for a meaningful outcome. The authors found that the total uncertainty for the OClO data sets investigated in this study ranges from about 26% to 33% for the different stations. They furthermore found that satellite and ground-based data sets show a good agreement for the inter-annual variability and the overall seasonal behaviour at the different sites. But they also found a median bias of about $-2.2x10^{13}$ molec/cm2 over all stations for both GOME-2 instruments with individual biases up to $8x10^{13}$ molec/cm2.
The validation study is comprehensible and clearly presented in the manuscript, and the authors also provide a more in-depth description of the three sensitivity tests in Appendix A2. The paper is definitely recommended for publication in AMT after the specific comments below have been addressed.

*Answer: We thank the reviewer for his useful comments and suggestions. We answer to each point below.*

Specific comments:
Page 1, line 1: '... produced within the ...' *-->done*
Page 1, line 3: Only measurements up to 2016 are discussed in this paper. Why was this study not extended to include at least some data from the most recent 5 years (2017 – 2021)?
Unfortunately, these GOME-2 data are from a data record that has only been produced until 2016 (see https://acsaf.org/datarecords/oclo_vcd.php) and therefore the corresponding data for the 2017-2021 period do not exist. The following modification has been included in the text:
"using the GOME2-A and -B instruments measurements covering over the 2007-2016 and 2013-2016 periods, respectively."

Page 1, lines 6-13: The uncertainty for the ground-based data sets is provided in the abstract as a percentage (lines 6/7) while the bias between ground-based and satellite data is given as an absolute number (lines 11/13). It would certainly be helpful if one of the two quantities could be provided as both, percentage and absolute value. That would make it easier to understand and interpret the information provided in the abstract, and it would put the retrieved bias and the uncertainty into context.
We thank the reviewer for this comment. The absolute uncertainty values are now also added in the abstract. As discussed in Sect 3.1.1 and Table 3, the random uncertainties are estimated for an SCD of around 15e13 molec/cm², so this SCD value is used also for the conversion from relative to absolute values of the other uncertainty sources. Therefore 25% maximum systematic uncertainty corresponds to 3.75 e13 molec/cm², and 26% to 33% total uncertainties correspond to about 4-5e13 molec/cm². The 4% to 16% expected systematic bias against GOME-2 correspond to about 0.6 to 2.4e13 molec/cm². These values have also been added in Sect 3.1.1.

Page 1, line 7: '... data analyses ...' *-->done*
Page 2, line 19: '...associated with strong ...' ' *-->done*
Page 2, line 35: '...its Amendments.' ' *-->done*
Page 3, line 51: delete 'study' ' *-->done*

Page 3, line57: '… mostly for a few … ' *-->done*
Page 3, line 59: 'In this paper, …' *-->done*
Page 3, line 60: Add space between 'AC' and 'SAF' *-->done*
Page 3, line 63: '… comparison method.' *-->done*
Page 4, line 92: Replace comma with space after 'orbit' *-->done*
Page 4, lines 90-93: Would be great, if you could give the reader an idea here regarding how big the amount in this bias correction is compared to actual OClO amount? E.g. how does this amount compare with the median bias quoted in the abstract.
This can be a rather important normalization/offset correction, which is needed because there can be (large) biases between the OClO SCDs from orbit to orbit (e.g. when the solar reference spectrum changes).
Typically, the offset can be can be a few (~1-4) e13 molec/cm2, but it seems to be corrected very well since it a systematic bias in the SCD on top of the OClO signal. Since there is no OClO at the lower latitudes, the large systematic bias can be accurately removed by this offset correction.
The following sentence has been added in the manuscript:
Typically, the offset can be can be a few (~1-4) e13 molec/cm2.

Page 5, Figure 1: It would really help with the readability of the plot if the text and legend would be bigger. Also add 'SCD' after 'OClO' in the caption.
The figure has been modified as suggested.

Page 5, line 96: Add comma after 'circumstances' *-->done*
Page 5, lines 103 & 105 & 107: Capitalize 'Hemisphere' when its used in combination with 'Southern' or 'Northern'. *-->done*
Page 5, lines 106 – 108: Not sure if I quite follow this interpretation here. For me, it looks more like GOME-2A for the NH starts with a baseline close to 0 for the first 3 years, then has a jump up in 2010 before it slowly drifts down again to a 0 baseline in 2016. GOME2-A for the SH starts negative, drifts up until in it is in the positive in 2010/2011, but then jumps straight down again in 2011/12 and stays in the negative.
Based on this comment, the discussion is now extended as follows:
"This is partly the case in the first years of measurements of each instrument, especially in the Northern Hemisphere, although some negative or positive offsets (of up to 4 to 5 x10$^{13}$ molec/cm²) and drifts appear for some of the years (e.g. 2010 in the Northern Hemisphere for GOME-2A). In particular, GOME-2A for the Northern Hemisphere starts with a baseline close to 0 for the first 3 years, then has a jump up in 2010 before it slowly drifts down again to a 0 baseline in 2016. For the Southern Hemisphere GOME-2A starts negative, drifts up until it is in the positive in 2010/2011, and then jumps straight down again in 2011/12 and stays in the negative."

Page 6, line 127: comma after 'From Table 2' *-->done*
Page 7, Figure 4 & Fig 4 caption: I like Figure 4, it's a nice visualisation of the different wavelength intervals used. To figure out which interval is used by which group, this can be identified via Table 2. Just to make it a bit easier, would it be possible to add the group names into Fig 4 straight behind the wavelength interval? Or alternatively, the group names could also be added in the caption e.g. in the order of appearance from top to bottom.
We thank the reviewer and we followed his suggestion of adding the group names in the caption of Fig. 4.

Page 7, Fig 4 caption: add 'analysis' after 'DOAS', just to clarify that this is not the wavelength interval each instrument covers but the interval each group uses for their data analysis. *-->done*

Page 7, line 138: 'Also, ...' -->*done*
Page 8, line 139: Add comma after 'needed' -->*done*
Page 8, line 141: Just to be clear, water vapour should have been included but it was not, correct? Could clarify that in the text.
The NIWA analysis has considered water wapor for the OClO retrieval. We changed the "'should be" to "is" to clarify.

Page 8, line 146: 'In this section, ...' -->*done*
Page 9, line 160-161: Is there any particular reason why Ny- Ålesund was chosen to be the test site?
Ny-Ålesund was used as a test case because (1) there was a close collaboration with the IUPB group for this OClO work and (2) the spectra are known as being of good quality. Spectra from the BIRA Harestua instrument could not be used as the spectral coverage was smaller (only up to 379nm, see table A1).

Page 9, lines 168: How were the median OClO SCD values determined, e.g. were any selection criteria applied?
The median OClO SCD values are, for each spectra, the median values of the OClO SCD retrieved with the different group's choices/cases. There were no specific selection criteria applied.

Page 10, line 192: '... lead to a systematic ...' -->*done*
Page 10, line 205: '... used as input for...' -->*done*
Page 11, line 218: '... measurements at Arrival Heights. At this site, the ...' -->*done*
Page 12, Figure 7 caption: '... the offset ...' and delete 'of' before 'Neumayer' -->*done*
Page 13, line 238: delete the 2nd 'et al.' -->*done*
Page 13, line 242 – 244: The authors state: 'On average, over the 85° to 92° SZA range, the AMF difference is close to zero.' However, looking at Figure 8, this is still between 5% and -8% ... is that accounted for?
The 5 to -8% difference on the AMF dependence on SZA is not taken into account in the present work, and could therefore explain part of the remaining differences between GOME-2 and ground-based SCDs. Fig. 17 shows that generally SCD_Sat< SCD_gb, for valid flags (ie >85°SZA), but this could be compensated in the VCD by the AMF. We should thus also have AMF_sat < AMF_gb, but Fig. 8 shows that this is only the case for SZA>88°. We added a comment in this sense also in Sect. 4.3 when also discussing potential explanations of the remaining differences, following reviewer 2 request.
"..). The impact of the AMF differences highlighted in Fig. 8 is also a multiplicative effect. The smaller satellite SCDs for valid flags (ie >85°SZA) found here compared to the ground-based ones, could be potentially compensated in the VCD by the AMF. Fig. 8 shows that AMF_sat is smaller than AMF_gb, only for SZA>88°. "

Page 14, line 247: '... OClO SCD measurements...' -->*done*
Page 14, line 252: '... mid-May...' -->*done*
Page 14, line 254: '... is larger in ...' -->*done*
Page 14, line 255: '...OClO SCDs...' -->*done*
Page 14, line 260: 'At Arrival Heights, ...' -->*done*
Page 14, line 262: '... mid-April...' -->*done*
Page 14, line 264: '...overpasses are ...' -->*done*
Page 14, line 266: 'Each year, ...' -->*done*
Page 15, Fig 9 & page 16, Fig 10 captions: '...there are no ...' Same also for Figures 13 & 14 -->*done*
Page 18, line 288: '... can only be made during April/May ...' -->*done*
Page 18, line 291: '... SCDs ...' -->*done*

Page 18, line 299: '… prevent detection of the other …' *-->done*
Page 19, line 300: 'The large OClO peak at Ny- Ålesund and Kiruna in early 2008 …' (just to be clear) *--
>done*
Page 20, line 304: Should it be 'over Ny- Ålesund and Kiruna' rather ? *-->yes, thanks!*
Page 20, line 309: Shouldn't that be 'GOME2-A SZA'? *-->actually, this is the case for both GOME-2A and −
B sensors. This has been specified in the text.*
Page 20, line 315: Add 'respectively' in the bracket *-->done*
Page 20, Figure 15: Would be interesting to have the same plot for one more station, in particular e.g. for
Ny- Ålesund (NH station).
The figure for Ny-Ålesund is included below. We however think it does not bring so much to the
discussion, as the number of points for Ny-Ålesund during the first years of operation is quite small (from
534 to 191 points). However, the improvements in the comparison are clear: reduction in RMS (from
3.5e13 to 2.7e13), almost half the value of the offset and increase of the slope (from 0.87 to 0.91).

[Figure]

Page 22, Fig 16 caption: '… defined as follows:…' *-->done*
Page 23, Fig 17 caption: '… during the active months.' *-->done*
Page 24, line 359: 'For the ground-based …' *-->done*
Page 25, line 377: Replace 'points' with something like 'measurements' or 'data' *-->done*
Page 25, line 380: 'OClO GOME2 products …' *-->done*
Page 25, line 379-381: On what study or analysis is the conclusion based that the GOME2 OClO data
product discussed within this manuscript meets the AC SAF mission requirements? Either this needs to be
explained in more detail in the text or the relevant reference together with a short summary needs to be
provided.
A sentence making the link between the different hemispheric biases found in this study and the AC SAF
mission requirements has been added at the end of Sect. 4.3 and the reference to the AC SAF mission
requirement document (Hovila and Hassinen, 2021) is also added here.
"These numbers are within the EUMETSAT AC SAF GDP OClO product target accuracy of 50% and close to
the optimal accuracy of 30% (Hovila and Hassinen, 2021)."

Page 25, line 390: comma after 'retrievals' *-->done*
Page 24, line 401: 'At the end of 2012, a new instrument was installed …' *-->done*

Page 24, line 405: '… since 1999 ….' (delete 'the') -->*done*

Page 26, line 406: 'Generally, …' -->*done*

Page 26, lines 412 & 418: '… a Vis zenith-sky DOAS at …' (more consistent with the rest of the text) -->*done*

Page 26, line 414: '… during the winter/spring season …' -->*done*

Page 26, lines 415 & 419: '… UV/Vis MAX-DOAS was…' -->*done*

Page 26, lines 415 – 417: The last 2 sentences in this paragraph should be switched around. -->*done*

Page 26, lines 415 & 421: 'OClO SCDs' -->*done*

Page 26, line 416: 'Ground-based SCD measurements…' (delete s in SCDs) -->*done*

Page 27, line 424: '… OClO SCD analyses …' and looks like a bracket is missing after 'window' -->*done*

Page 27, line 429: '… what is used for …' -->*done*

Page 27, line 433: '… each group's OClO cross-section …' -->*done*

Page 27, line 440: '… each group's choice …' -->*done*

Page 27, line 443: '… the Ny-Ålesund …' ' -->*done*

Page 27, line 452: '… each group's analysis …' ' -->*done*

Page 28, Figure A1, caption: '… Ny-Ålesund … result on the cross-sections … DOAS analyses …. what is described in …' -->*done*

Page 29, Figure A2 & Figure A3, caption: '… OClO SCDs …' and '… Ny-Ålesund …'

Page 29, Figure A3, caption: '… DOAS analyses used ….' -->*done*

---

## Author Comment (AC2)

Reviewer comments are given in black and author answers are in blue. Changes in the revised manuscript are marked in red.

This manuscript describes comparisons between satellite-derived and ground-based observations of OClO. The manuscript is generally well written and presents important validation for the GOME-2 (A and B) satellite data records. The manuscript is well within the scope of AMT and with revision should be acceptable for publication. There are two areas in which the article should be clarified:

1) While the comparison is reported as a bias (offset between satellite observation and ground-based observation), the data appear to fit a model where the satellite observations have a lower slope and relatively small intercept. Potential reasons for this behavior should be discussed.

2) Some of the airmass factor analysis was not clear and needed further explanation.

*Answer: We thank the reviewer for his useful comments and suggestions. The main comments and specific issues are addressed point by point below.*

Concerning the remaining differences between satellite and ground-based, we agree that it can be fitted by a slope smaller than unity, with a relatively small intercept. The term "bias" to express the remaining satellite minus ground-based differences is thus maybe not the most appropriate, as the intercept is generally also referred as additive bias, compared to the multiplicative bias coming from the slope. We however do not agree that the comparison is only discussed in term of a bias. In Sect. 4.3 and Table 4 the results are discussed both as a bias/offset (fig. 16) and as slope and intercept (fig 17). We changed the term bias to "SAT-GB offset" at the end of the abstract (lines 11-13) to minimize the confusion. The small intercept can potentially be explained by the GOME-2 normalization correction (see Sect. 2, lines 88-93), that subtracts any remaining positive OClO SCD in region where no OClO is expected. The slope smaller than one can potentially be explained by the differences between the GOME-2 and ground-based DOAS settings and the corresponding SCD uncertainties. For GOME-2 there is e.g. the impact of the mean residual or the scan angle empirical correction functions, see Sect. 2 lines 84 to 88, while the uncertainties from the ground-based data is estimated to be between 26% to 33%, which is close to the remaining multiplicative bias from the slope (0.64 and 0.72 for GOME-2A and GOME-2B, respectively). This discussion has been further extended in Sect. 4.3 (details below, in the comment about Table 4).

Concerning the airmass factor analysis, we have redone Fig. 8 with the same convention than in Fig. 6 for the sake of consistency and clarity of the discussion (see further explanations below).

Specific issues (listed by line number) are below:

Abstract, line 10: Are these slopes compared to ground-based observations?

Yes, these are slopes compared to the ground-based observations. This has been specified in the abstract

now. This has been clarified as in the text (line 11) as follows:
" with slopes of 0.64 and 0.72 with respect to the ground-based data ensemble, respectively."

Line 78: This appears to be an anonymous FTP server.  Many of these servers will transition to other more secure means.  Is this transition envisioned?  How will data be accessed in the future?
Besides sftp and ftps, anonymous FTP is also a relative secure (and user friendly) option, and there are no short term plans to change it.

Line 84: Is "Eta" explained in some citable source (e.g. the ATBD?). If so, please cite it.
The citation to the ATBD has been added.

Line 90: This "normalization" seems more like a "bias" correction for the orbit based upon a region where OClO is not expected.  Should this be called a bias correction?
We agree with the reviewer that this normalization is a sort of bias correction. But the term "normalization" is used in the algorithm ATBD (Valks et al., 2019), so we would like to keep the same naming convention.

Line 116: is "Annex A1" the right name?  Appendix?
We agree "Appendix" is more appropriate. This has been changed here, and in other places of the manuscript (e.g. in Section 3.1.1, line 171)

Line 153: I am unclear on how the errors are being represented.  If I understand right, you are expressing a percentage of 15 x 10^13molec/cm^2.  Doesn't that mean you are really talking about a SCD error (say 10% of this "reference" would mean 1.5 x 10^13 molec/cm^2.).  I think that if the primary values are SCD errors, they should be expressed that way, and you can then give the percentage of this reference as secondary.  Note that the 15+/-2 is quite confusing in this context.  Are you dividing by a number with error and also propagating this error?
We reported SCD errors as the median value of the DOAS fit errors for SCD values in the 13 to 17x 10^13 molec/cm^2 range (i.e., 15 plus or minus 2 x 10^13molec/cm^2). This is done because some stations have much higher or lower SCD values range, and we wanted to report the error using a "common baseline". We then report the random error as a percentage value of 15x10^13 molec/cm^2 to be coherent with the systematic uncertainty estimates, provided as a percentage.
Also, to address a comment from reviewer 1, the absolute values of the SCD errors are now also given in Sect 1.1.3 and in the abstract.

Line 171: Typically bias is an offset in intercept, and slope error is a multiplicative error.  I think you mean that the intercepts are small, so the differences between the measurements are mostly in the slope.
This is indeed what we meant. The sentence has been partly revised to clarify:
All intercepts except for IUPH are small (smaller than $1x10^{13}$molec/cm$^2$ (see Fig. A2)), and the differences between the measurements reside mostly in the slope, meaning that those differences are mostly multiplicative.

Line 178-185: I don't understand this; it is exploring the different cross sections?  How is this done?
The third test is performed with a similar methodology  than the second one, ie performing  regression analyses of OClO SCD retrieved with different settings (different y-values, see the last line of table 3)

with respect to one set of OClO SCD (x-values). In this case, the x-values in figure A3, are the SCDs obtained from the ground-based Ny-Alesund spectra using the GOME-2 settings instead of the median OClO SCD values (as in fig. A2). As for the second case, the effect of the possible different OClO cross-sections is further added as a separate step in table 3.

The first sentence in the paragraph has been revised to:

"The expected systematic bias due to differences between each group's analysis and the GOME-2 OClO retrieval settings is investigated in a third test. This test (presented in Figure A3) uses a similar methodology than the second test presented above, but we now compare the SCDs obtained by applying to the Ny-Alesund spectra the DOAS settings from the different groups and the GOME-2 settings defined in Table 1. "

Table 3: What is "-" and are "n.a." in this table?  If the "-" means no error, why not say 0?  Ny-Alesund seems to also use the 213K OClO cross section, wouldn't that lead to no error, but 2.5 is listed.

"-" means indeed zero. This has been changed in Table3 for the revised manuscript. "n.a" means non applicable. This is the case for the Arrival Height NIWA measurements which have a specific wavelength range (in the visible) outside the range of the Ny-Alesund spectra. Therefore, this case could not be tested, as explained in the text.

The Ny-Alesund test case is a bit peculiar regarding the OClO cross-section. The Kromminga et al. cross-section at 213K was used, but it is an older reference (from 1999 instead of 2003, see line 134) and tests have shown that there is a bias between both cross section sources of about 2.5%. The "b" footnote of Table 2 has been put in bold, as done for other information, when being different than the OClO cross-section baseline.

Line 200-202: How does this AMF relate to "photochemical AMFs"?  It seems like Figure 7 lower panel indicates that the OClO SCD is within noise of zero at 70 degrees?  Can this be explained further?

In Figure 7, the whole time-series is presented (irrespective of whether chlorine activation takes place or not), so, on average over all the years, the OClO SCD is indeed around zero at 70°. The correction itself is calculated and applied to each individual day. On activated days, such as illustrated in Fig. 5, the OClO SCD is non zero at 70° SZA. In such conditions, measured SCDs follow the same behavior as the photochemical AMF (as demonstrated by the linearity of the Langley plot in Fig. 5). Note that the OClO AMF used in Fig. 5 corresponds to the median photochemical AMF displayed in red in Fig. 7.

Figure 7 caption typo -- it says "offsset" --> corrected, thanks.

Figure 8 shows AMFs that are quite different from the AMFs shown in Figure 6.  Can it be explained why these two AMFs are so different and how both are used?

The AMFs in both figures are defined differently. In figure 6, the AMF is defined with respect to the vertical column at 70° SZA, while in figure 8 the AMF is defined relative to the VCD at each SZA (i.e. a different VCD value for each angle). In the first case (Fig. 6), the AMF is proportional to the measured SCDs and therefore easier to interpret. To avoid any confusion, we decided to redo figure 8 using the same definition as in figure 6. We had to change date and latitude as on the 10th of February at 65°N, the minimum SZA of the day is of about 77°, and therefore 70° is not reached. The new day (01/03/2000) corresponds to the high activation case (blue line) of figure 6.

Note that the AMF of Figure 8 is not used in the calculation and its purpose is to provide an illustration of the difference between the AMF in both satellite nadir and ground-based zenith geometries.

The previous figure 8 is thus replaced in the manuscript by the following one:

[Figure]

Figure 8. OClO AMF calculations for 60°N from ground-based zenith and satellite nadir geometries

On Figure 9, some points seem to go below the lowest values plotted on the plot.  This seems particularly true of the Neumayer data.

This is indeed true, mostly for Neumayer, where there are several points down to -3.5e14 molec/cm². There are about 27 points smaller than -1e14 molec/cm² over 1536 total GOME-2A satellite comparison points (and 5 over 633 for GOME-2B), see figure below with time-series and scatter plot.

However, we think that in the interest of displaying (1) as many details as possible and (2) maintaining a coherence between both hemispheres and between GOME-2A and GOME-2B, it is better not to redo all plots of figures 9 to 14 with a smaller limit on the x-axis.

We have added a warning in the caption on figure 9 :

"Please note that in some cases, some GOME-2A points lie below the x-axis limit of -1e14, down to -3.5e14 molec/cm², especially from 2011 onward (e.g., in the case of Neumayer, this represents 27 data points over a total of 1536)."

and in the text:

"It can be noted that in the case of GOME-2A, some daily mean points are negative and smaller than the lower x-axis limit in Fig. 9. This is especially the case from 2011 on, when data are more negative, as also seen in Fig. 2 and discussed at the end of Sect. 2."

[Figure]

Line 298-299: Can the authors explain the sentence "Unfortunately the gap in ..." I don't understand the information in the parentheses about "...pixels SZA..." or the "...prevents to detect the..." phrases.

Actually, the GOME-2 instruments only retrieve valid OClO SCDs in a SZA range between 85° and 92°, as explained in lines 99-101 and 228 and this has an impact on the possible GOME-2 OClO measurements (see lines 232-235). Between February and May, the SZA around some of the northern hemispheric sites is systematically smaller than 85°, and thus no valid OClO SCD can be retrieved, while ground-based instruments continue to measure OClO at twilight. An illustration of the GOME-2A SZA variability around two illustrative sites is given in the figure below, where the reduction of the number of measurement points after July 2013 and the change in swath configuration, are also seen.

[Figure]

Illustration of the SZA time evolution for all the pixels within a radius of 200 km around the station. In July 2013 GOME-2A swath is reduced to half of his nominal length, which impacts the coverage around the stations. The black dotted line corresponds to 85° SZA.

The sentence has been revised to "Due to the low SZA values (systematically smaller than 85°SZA) around the sites between February and May, no valid OClO SCDs could be retrieved by GOME-2, while some OClO activation peaks are detected during this period by the ground-based instruments measuring at twilight."

Table 4 and discussion of Table 4: Although there is a low bias, it appears that the intercepts are a lot smaller than the bias, indicating that the slope being under unity is the largest contributor to the negative bias.

We agree with the reviewer, and some more discussion in this sense is added in Section 4.3:

"The small intercepts are representative of small additive biases, while the slopes smaller than unity are the largest contributors to the negative multiplicative bias.  The small intercept can potentially be explained by the GOME-2 normalization correction (see Sect. 2), that subtracts any remaining positive OClO SCD in region where no OClO is expected. The slope can potentially be explained by the different GOME-2 and ground-based DOAS fit settings and the corresponding SCD uncertainties (see Sect. 3.1.1). For GOME-2 there is e.g. the impact of the mean residual or the scan angle empirical correction functions (see Sect.2). The impact of the AMF differences highlighted in Fig. 8 has also a multiplicative effect. The smaller satellite SCDs for valid flags (ie >85°SZA) found here compared to the ground-based ones, could be potentially compensated in the VCD by the AMF. Fig. 8 shows that AMF_sat is smaller than AMF_gb, only for SZA>88°."
and
"due to their DOAS settings choices, and in general, there is a total uncertainty within the ground-based datasets of about 26 to 33%, which is close to the remaining 36 and 28% multiplicative biases from the slope (slope values of 0.64 and 0.72 for GOME-2A and GOME-2B respectively). "

Line 350: Day-to-day variations are mentioned in OClO, but it is a bit hard to see on the plots that the data follow on these timescales.  It appears from the data that many of the variations are on a slower than day-to-day timescale, so it seems like the agreement is more in the longer term behavior.
We partially agree that it is hard to see on those figures the day-to-day variability since OClO variations are usually on a longer time scale. There are however, some short-time changes (day-to-day), and these are usually seen by both ground-based and satellite measurements. This is why we used the 'day-to-day variation' formulation. The number of comparison data points per month oscillates between zero to around a dozen on average and can reach up to almost 30 on some months. Two examples of short-term variations (zooms of Fig. 9 and 10) observed at Arrival Height and Neumayer are presented in the figure below. The numbers in the table below correspond to the number of comparison data points for the selected months at both stations.

[Figure]

[Figure]

| | |
|---|---|
| Neumayer, 2015 1: 25 points | ArrivalHeights, 2012 1: 1 |
| Neumayer, 2015 2: 14 | ArrivalHeights, 2012 2: 20 |
| Neumayer, 2015 3: 25 | ArrivalHeights, 2012 3: 21 |
| Neumayer, 2015 4: 25 | ArrivalHeights, 2012 4: 9 |
| Neumayer, 2015 5: 2 | ArrivalHeights, 2012 5: 0 |
| Neumayer, 2015 6: 0 | ArrivalHeights, 2012 6: 0 |
| Neumayer, 2015 7: 0 | ArrivalHeights, 2012 7: 0 |
| Neumayer, 2015 8: 12 | ArrivalHeights, 2012 8: 3 |
| Neumayer, 2015 9: 13 | ArrivalHeights, 2012 9: 26 |
| Neumayer, 2015 10: 11 | ArrivalHeights, 2012 10: 21 |
| Neumayer, 2015 11: 29 | ArrivalHeights, 2012 11: 11 |
| Neumayer, 2015 12: 30 | ArrivalHeights, 2012 12: 0 |

In a number of places near Figure A.3 caption, "Ny-Alesund" is misspelled. --> corrected, thanks.